# Distinctive mechanisms of epilepsy-causing mutants discovered by measuring S4 movement in KCNQ2 channels

**Michaela A Edmond[1], Andy Hinojo-Perez[1], Xiaoan Wu[2], Marta E Perez Rodriguez[2], Rene Barro-Soria[1]***

[1]Department of Medicine, Miller School of Medicine, University of Miami, Miami, United States; [2]Department of Physiology and Biophysics, Miller School of Medicine, University of Miami, Miami, United States

**Abstract** Neuronal KCNQ channels mediate the M-current, a key regulator of membrane excitability in the central and peripheral nervous systems. Mutations in KCNQ2 channels cause severe neurodevelopmental disorders, including epileptic encephalopathies. However, the impact that different mutations have on channel function remains poorly defined, largely because of our limited understanding of the voltage-sensing mechanisms that trigger channel gating. Here, we define the parameters of voltage sensor movements in wt-KCNQ2 and channels bearing epilepsy-associated mutations using cysteine accessibility and voltage clamp fluorometry (VCF). Cysteine modification reveals that a stretch of eight to nine amino acids in the S4 becomes exposed upon voltage sensing domain activation of KCNQ2 channels. VCF shows that the voltage dependence and the time course of S4 movement and channel opening/closing closely correlate. VCF reveals different mechanisms by which different epilepsy-associated mutations affect KCNQ2 channel voltage-dependent gating. This study provides insight into KCNQ2 channel function, which will aid in uncovering the mechanisms underlying channelopathies.

*For correspondence:
rbarro@miami.edu

**Competing interest:** The authors declare that no competing interests exist.

## Editor's evaluation

This study makes an important technical advance with measurements of voltage-dependent conformational changes of KCNQ2/Kv7.2 channels, measurements which are known to be extremely difficult for this biologically important channel. This advance sheds light on the mechanism by which two human mutations act and opens the door to further investigations of voltage sensor movement in these channels.

## Introduction

Voltage-gated K+ (Kv) channels play a crucial role in regulating excitability, and dysregulation of their function has been associated with several disorders, including cardiac arrhythmias, epilepsy, and autism. Kv channels, including all members of the Kv7 family (Kv7.1–Kv7.5, also known as KCNQ as they are encoded by KCNQ1–5 genes; *Jespersen et al., 2005*; *Jentsch, 2000*; *Abbott and Pitt, 2014*), are highly heterogenous and widely expressed in excitable cells where they regulate resting membrane potential, shape the firing and duration of action potentials, and control rhythmic events (*Hille, 2001*).

One of the major potassium currents throughout the central and peripheral nervous systems is the M-current. The M-current is mainly conducted by heterotetrameric combinations of KCNQ2/3 and KCNQ3/5 channels (*Brown and Adams, 1980*; *Halliwell and Adams, 1982*; *Wang et al., 1998*;

*Schroeder et al., 2000*), but homotetrameric assemblies of channel subunits have also been shown to generate the M-current in neurons (*Schroeder et al., 2000*; *Soh et al., 2014*). KCNQ are non-inactivating channels with slowly activating and deactivating kinetics and a negative voltage for half-activation (*Brown and Adams, 1980*; *Halliwell and Adams, 1982*). These biophysical properties make KCNQ channels important regulators of neuronal excitability. For example, the peculiar lack of inactivation at voltages near the threshold for action potential initiation confers KCNQ channel's dominant role in regulating membrane excitability as one of the main outward sustained currents. Thus, inhibition of the KCNQ channel lowers the action potential threshold and slows excitatory post-synaptic potentials (*Adams et al., 1982*), resulting in reduced adaptation and prolonged repetitive neuronal firing (*Adams et al., 1986*). Among the KCNQ family of proteins, KCNQ2 channels have received particular attention because mutations in this channel have been associated with a variety of neurodevelopmental phenotypes (*Jentsch, 2000*; *Greene and Hoshi, 2017*; *Geisheker et al., 2017*), including epileptic encephalopathy (*Weckhuysen et al., 2012*; *Weckhuysen et al., 2013*; *Orhan et al., 2014*; *Saitsu et al., 2012*; *Kato et al., 2013*; *Rauch et al., 2012*), and more recently, autism (*Cornet et al., 2018*). Since KCNQ2 channels are central to physiological and pathophysiological events, it is important to understand the voltage-dependent mechanisms underlying channel opening and thereby define its role in physiological control of neuronal excitability, as well as providing a better understanding of how specific disease-associated variants alter KCNQ2 channel function.

The recently elucidated cryo-EM structure of human KCNQ2 channels (*Li et al., 2021b*) shows that, like canonical Kv channels (*Long et al., 2005*), KCNQ2 has a domain-swapped tetrameric architecture with six transmembrane helices (S1–S6) and cytosolically oriented N-terminal and C-terminal that form functional tetramers. The S5–S6 of the four subunits form a centrally located potassium selective pore that is flanked by the four voltage- sensing domains (VSDs; S1–S4; *Long et al., 2005*), where the C-terminal end of the S6 segments forms the gate (*Long et al., 2005*; *del Camino and Yellen, 2001*). Similar to that seen in other Kv channels, the fourth transmembrane segment contains several highly conserved positively charged amino acid residues that move in response to changes in membrane voltages that function as the voltage sensor (*Mannuzzu et al., 1996*; *Larsson et al., 1996*; *Seoh et al., 1996*; *Aggarwal and MacKinnon, 1996*). Gating current recordings have not been resolved for KCNQ2 channels, likely due to low channel density within the membrane and/or the slow kinetics of activation compared to other Kv channels (*Miceli et al., 2012*). Insight into the S4 movement of KCNQ2 has been inferred by previous mutagenesis studies showing that charge neutralization of the arginine residues in S4 altered the voltage sensitivity of channel opening (*Soldovieri et al., 2019*; *Miceli et al., 2008*). In addition, a disulfide crosslinking study showed that cysteine-substituted residues in the extracellular end of S4 crosslinked with a cysteine in S1 only in the closed state, further implying S4 movement (*Gourgy-Hacohen et al., 2014*). Although these studies provided insight into S4 rearrangements, they did not define parameters of S4 movement, such as the dynamic relationship between S4 activation and pore opening during voltage-controlled gating of KCNQ2 channels.

Our understanding of the voltage-controlled activation mechanisms of KCNQ2 channels is limited compared to other Kv channels like the related KCNQ1, whose physiological role in cardiac tissue has been extensively investigated (*Nerbonne and Kass, 2005*). Kv channel opening can occur either after all four S4 have been activated (*Zagotta et al., 1994*), or alternatively through independent activation of each S4 (*Horrigan et al., 1999*), as also reported for KCNQ1(35). Interestingly, pore opening of KCNQ1 channels can occur from two defined S4 conformations involving intermediate and fully activated S4 states (*Zaydman et al., 2014*; *Hou et al., 2017*; *Hou et al., 2020*). This activation scheme in which opening may occur from multiple S4 states, has provided a valuable framework to understand voltage-dependent gating of KCNQ1 with different accessory subunits, thereby allowing interpretation of its versatile physiology. The lack of mechanistic understanding of voltage-dependent gating in neuronal KCNQ2 channels has made it difficult to understand the impact that disease-associated variants will have on channel functionality. We here describe the mechanisms underlying voltage sensor movement in KCNQ2 channels relevant to understand epilepsy-associated KCNQ2 mutations.

We provide an extensive exploration of positions where cysteine could be inserted into the S3-voltage sensor (S4) loop and S4 helix and used with both voltage clamp fluorometry (VCF) (*Mannuzzu et al., 1996*) and cysteine accessibility (*Larsson et al., 1996*) to study S4 activation and its influence on disease-causing mutations. Cysteine accessibility reveals that a stretch of eight to nine S4 residues becomes exposed upon VSD activation of KCNQ2 channels. VCF shows that the voltage

dependence and the time course of S4 movement and channel opening/closing closely correlate. In addition, VCF data shows that two epilepsy-associated mutations – R198Q and R214W – perturb channel opening through two distinct mechanisms with R198Q directly altering S4 movement, while R214W uncouples voltage sensor movement and pore opening. These results provide critical information about KCNQ2 channel gating that will aid in future studies on KCNQ2 channelopathies.

## Results

### State-dependent external S4 modifications consistent with S4 as voltage sensor

The combination of cysteine-scanning mutagenesis and methanethiosulfonate (MTS) derivative modification is a powerful tool to study conformational changes in ion channel gating. This methodology assumes that covalent modification of substituted cysteines leads to functional changes in channel gating (*Figure 1A*). We test how the fourth transmembrane domain (S4) moves in the KCNQ2 channel by measuring state-dependent accessibility changes of introduced cysteines in the S4 (or in the S3–S4 linker) (*Figure 1A–B*). We assess the state-dependent modification of substituted cysteines by plotting the membrane-impermeable thiol reagents (MTS)-induced change in current against the cumulative exposure to MTS reagents at either hyperpolarized (closed) or depolarized (open) voltages (See Materials and methods section and voltage protocols on top of *Figure 1C*). This approach has been previously used to demonstrate that S4 crosses the membrane during gating of the Shaker channel (*Larsson et al., 1996*) and assumes that changes in state-dependent modification rates of substituted-cysteines by externally applied MTS compounds indicate that some residues in S4 move (outward) across the membrane during channel activation.

In total, we made eight cysteine mutants within the S4 (or in the S3–S4 linker) of KCNQ2 channels (*Figure 1* and *Figure 1—figure supplement 1*). The cysteine mutants (N190C, A193C, S195C, A196C, R198C, S199C, L200C, and F202C) vary in steady-state conductance/voltage curve (G(V)) when compared to wild-type-KCNQ2 channels (*Figure 1—figure supplement 1B*, and *Supplementary file 1*). We express these mutants, one at a time, in *Xenopus* oocytes and use two-electrode voltage clamp to probe the external accessibility of the substituted cysteines to the MTS reagent (2-[ammonium]ethyl) methanethiosulfonate (MTSET) at both hyperpolarized (closed) and depolarized (open) voltages (*Figure 1* and *Figure 1—figure supplement 1*). As an important control, MTSET does not modify wt-KCNQ2 at either depolarized or hyperpolarized voltages (*Figure 1—figure supplement 1A* and *Supplementary file 1*). Note that the perfusion system quickly delivers a 5 s pulse of MTSET to the external surface of oocytes. This ensures perfusion of MTSET only at the indicated voltage as shown by the time course of solution exchange from 100 mM NaCl to 100 mM KCl (*Figure 1—figure supplement 2*).

For each cysteine mutant, we measure a family of currents in response to 20 mV voltage steps from –140 mV to +40 mV before (*Figure 1C, E and G*, left panels, and *Figure 1—figure supplement 1C-G*, left) and after applications of MTSET in both states (*Figure 1C, E and G*, middle and right panels, and *Figure 1—figure supplement 1C-G*, middle and right panels). To assess the state-dependent modification of substituted cysteines, we first apply MTSET at hyperpolarized voltages (closed channels) for 5 s in between 25 s washouts for 8–15 cycles and assayed the change in current at +20 mV (*Figure 1C, E and G*, 'closed state'-middle panels). On the same cell and after MTSET is washed out of the bath, we repeat a similar protocol but now applying MTSET at +20 mV (*Figure 1C, E and G*, 'open state'-right panels). External application of MTSET in the closed state significantly increases the current amplitude and shifts the G(V) relationship of N190C channels to the left ($\Delta GV_{1/2\,N190C\,closed}$ = –6.3 ± 1.3 mV, n=11, *Figure 1C–D* and *Figure 1—figure supplement 3A-B*, gray). We find that after the second MTSET application (now using the open state protocol) there is no additional increase in the current amplitude, and the G(V) relationship is not shifted further ($\Delta GV_{1/2\,N190C\,open}$ = –7.0 ± 1.7 mV, n=9, *Figure 1C–D*, and *Figure 1—figure supplement 3A-B*, yellow), as if all N190C channels were fully modified in the closed state. To test whether N190C is also accessible in the open state, we performed a separate experiment in which MTSET is applied at +20 mV (*Figure 1—figure supplement 4*). Using this protocol, we find that MTSET also increases the current amplitude and shifts the G(V) relationship of N190C channels to negative voltages ($\Delta GV_{1/2\,N190C\,open}$ = –12.2 ± 10 mV, n=5, *Figure 1—figure*

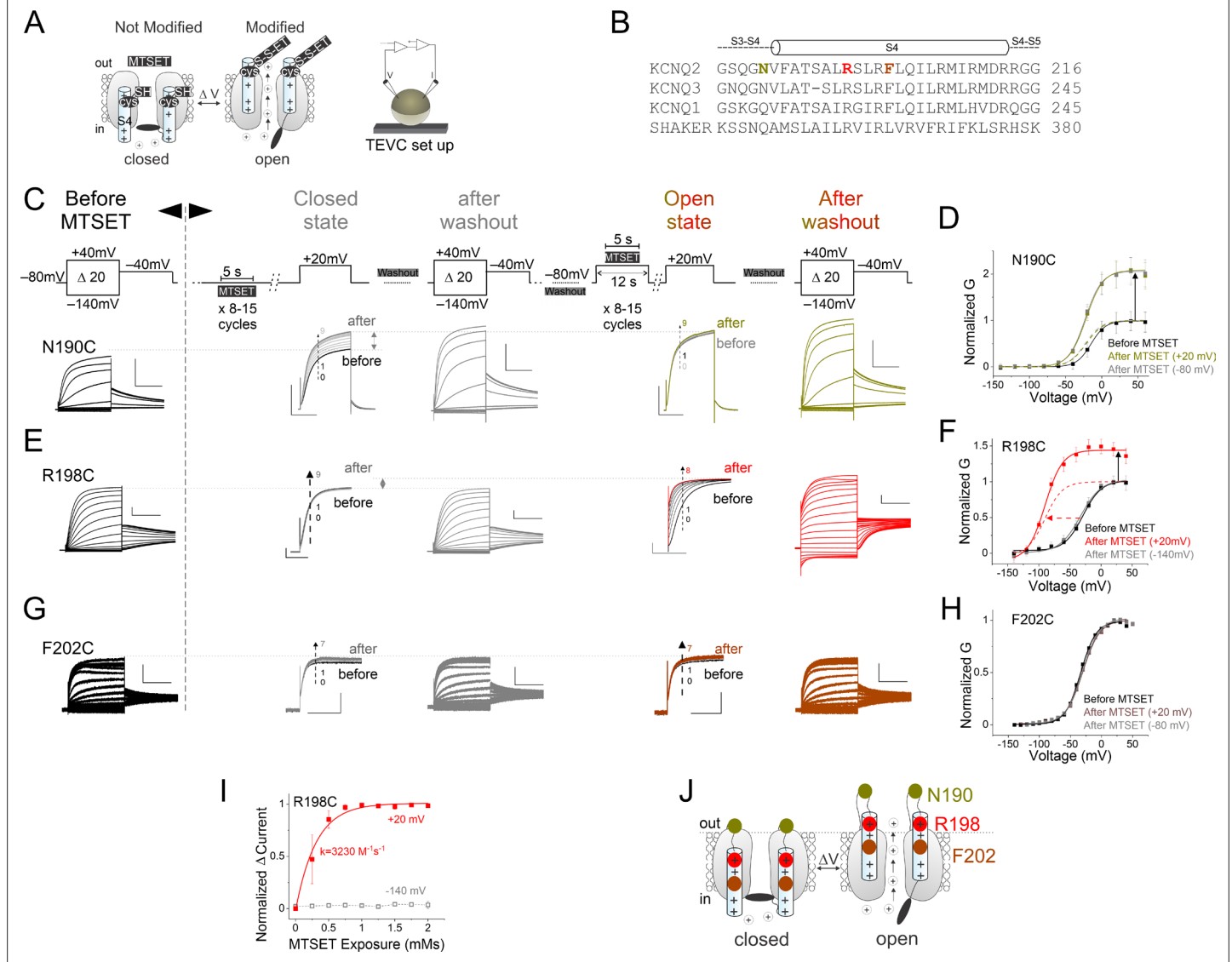

**Figure 1.** State-dependent modification of KCNQ2-R198C by external methanethiosulfonate (MTSET) is consistent with outward S4 motion. (**A**) Cartoons showing cysteine accessibility method with MTSET and two-electrode voltage clamp setup. (**B**) Sequence alignment of homologous S4 residues in KCNQ2, KCNQ3, KCNQ1, and Shaker channels. (**C, E, and G**) Currents from oocytes expressing (**C**) KCNQ2-N190C, (**E**) R198C, and (**G**) F202C channels in response to 20 mV voltage steps from −140 mV to +40 mV (left panels) before and after applications of MTSET (after washout, gray) in the closed and (after washout, color-coded) open states. MTSET is first applied ('closed state'-middle panels) at −80 mV for 5 s in between 25 s washouts for 8–15 cycles and the change in current is measured at +20 mV. On the same cell and after MTSET is washed out of the bath, MTSET is reapplied ('open state'-middle panels) at +20 mV using a similar protocol. We used 100 μM MTSET in (**C**) and (**G**), and 50 μM MTSET in (**E**). (**D, F, and H**) Steady-state conductance/voltage relationships, G(V)s, (lines from a Boltzmann fit) of (**D**) KCNQ2-N190C, (**F**) R198C, and (**H**) F202C channels normalized to peak conductance before MTSET application (black). The G(V) relationships after MTSET application in the closed (−80 mV, gray) and open (+20 mV, color-coded) states are obtained from recordings of panels (**C**), (**E**), and (**G**), ('closed- and open state'-middle panels, respectively); mean ± SEM, n=3–24. (**I**) The rate of MTSET modification of R198C channels at +20 mV (red squares) or −80 mV (gray squares) was measured using the difference in current amplitudes taken at 400 ms after the start of the +20 mV voltage step, vertical dashed arrows in (**E**) between the first sweep (before MTSET application, which is represented by #0 along the vertical dashed arrows in (**E**) and normalized to zero) and the subsequent sweeps (after several MTSET application which are represented by #1, 2, …8–9 along the vertical dashed arrows in (**E**)) from the 'closed-state and open-state'-middle panels. The normalized delta current amplitude was plotted versus the cumulative MTSET exposure and fitted with an exponential. The fitted second-order rate constant in the open state protocol is shown in red. $k_{open}$ = 3230 ± 3.8 $M^{-1}$ $s^{-1}$ (n=8). (**J**) Cartoon representing the voltage-dependent cysteine accessibility data. MTSET modifies N190 in both the closed and open states. While F202 remains unmodified in both states (seemingly buried in the membrane), R198 becomes accessible only in the open state. Dashed line indicates the proposed outer lipid bilayer boundary.

The online version of this article includes the following figure supplement(s) for figure 1:

*Figure 1 continued on next page*

*Figure 1 continued*

**Figure supplement 1.** State-dependent modification of S4 residues by external methanethiosulfonate (MTSET) consistent with outward S4 motion.

**Figure supplement 2.** Fast perfusion system delivers 5 s applications of external solution exchange to whole oocytes.

**Figure supplement 3.** Summary of modification of N190C, R198C, and F202C in the closed and open states by external methanethiosulfonate (MTSET).

**Figure supplement 4.** Modification of N190C in the open state by external methanethiosulfonate (MTSET).

**Figure supplement 5.** Proposed molecular motions of S4 residues in KCNQ2 channels.

*supplement 4*). Together, these results suggest that N190 is accessible and exposed to the extracellular solution in both the closed and open states (*Figure 1J*, yellow).

For R198C, external MTSET application in the open state (at +20 mV), increases the current amplitude and left-shifts the G(V) relationship (*Figure 1E–F* red and *Figure 1—figure supplement 3C-D*, red). In contrast, when MTSET is applied in the closed state (at –140 mV), R198C channels are not modified (*Figure 1E–F* gray and *Figure 1—figure supplement 3C-D*, gray). Since MTSET modifies residue R198C relatively fast at depolarized potentials (*Figure 1E,I,* red) but not significantly at hyperpolarized potentials (*Figure 1E,I,* gray), that suggests that this residue is not accessible (i.e. is buried in the membrane) in the closed state (with S4 down) but becomes accessible in the open state (with S4 up, *Figure 1J*, red). We find similar state-dependent modifications upon external MTSET perfusion for KCNQ2 channels with cysteine substitutions at residues A193C, S195C, A196C, S199C, and L200C (*Figure 1—figure supplement 1C-G*). External application of 0.1 mM MTSET (and even up to 1 mM) shows no modification of channels with cysteines introduced further toward the C-terminus of the S4, such as F202C in either the open or closed states (*Figure 1G–H* and *Figure 1—figure supplement 3E-F*). F202 is the outermost N-terminal residue in the S4 segment to remain unmodified by external MTSET. This result suggests that either F202C remains buried in the membrane during S4 activation (unmodified), even under conditions of high MTSET concentrations and strong depolarization to +20 mV (*Figure 1J*, brown), or alternatively that the modification does not significantly alter channel gating. *Figure 1J*, *Figure 1—figure supplement 1H*, and *Figure 1—figure supplement 5* show cartoons summarizing a map of the voltage-dependent distribution of S4 residues in the resting and activated conformations inferred from the extracellular cysteine accessibility data.

## Tracking S4 movement of KCNQ2 channels using voltage-clamp fluorometry

VCF allows simultaneous measurements of S4 movement (by fluorescence) and gate opening (by ionic current) based on the physicochemical properties of fluorescent probes in different environments and on their short half-life once excited (*Lakowicz, 2006*; *Mannuzzu et al., 1996*). To identify the best candidate site for fluorescent tracking of S4 movement in KCNQ2 channels using VCF, we first performed, one at a time, cysteine substitution of residues in the extracellular S3-S4 linker (*Figure 2A*). We find that this region exhibits sensitivity to cysteine mutations (*Figure 2B–D*), similar to a previous report for homologous cysteine mutations in KCNQ3 channels (*Kim et al., 2017*). Compared to wt-KCNQ2 channels, the mutants Q188C, G189C, and N190C shift the steady-state conductance/voltage curve, G(V), toward positive voltages ($\Delta GV_{1/2}$ = +9.3 ± 0.7 mV, $\Delta GV_{1/2}$ = +24.2 ±0.7 mV, and $\Delta GV_{1/2}$ = +29.8 ± 0.3 mV, respectively), whereas the mutants V191C and F192C shift the G(V) curves toward negative voltages ($\Delta GV_{1/2}$ = –2 mV ± 0.9 mV and $\Delta GV_{1/2}$ = –12.5 mV ± 1.7 mV, respectively; *Figure 2C–D*, open symbols and *Supplementary file 1*). Unlike the F192C mutant, the wt channels and the other cysteine mutants exhibit a sigmoidal time course and appear to have multiple exponential components (*Figure 2B*), with the F192C mutant generating the fastest time course of current activation (*Figure 2—figure supplement 1A*). Moreover, all five cysteine substitutions showed a further leftward G(V) shift upon fluorophore labeling (*Figure 2D*, filled symbols). The mechanisms by which the cysteine substitutions and their dye-conjugated versions may alter some of the gating properties are unknown and were not investigated further.

Out of the five KCNQ2 substituted cysteines in the S3–S4 linker, the labeled mutant KCNQ2-F192C exhibits the most reliable and robust voltage-dependent fluorescence signals (maximum fluorescence change, dF/F~1%) that saturates well at negative and positive voltages, when either labeled with Alexa488 5-maleimide (*Figure 2E–G*) or DyLight488-maleimide (*Figure 2—figure supplement 1B,D*). The time courses of fluorescence signal labeled with either fluorophore are similar (*Figure 2—figure*

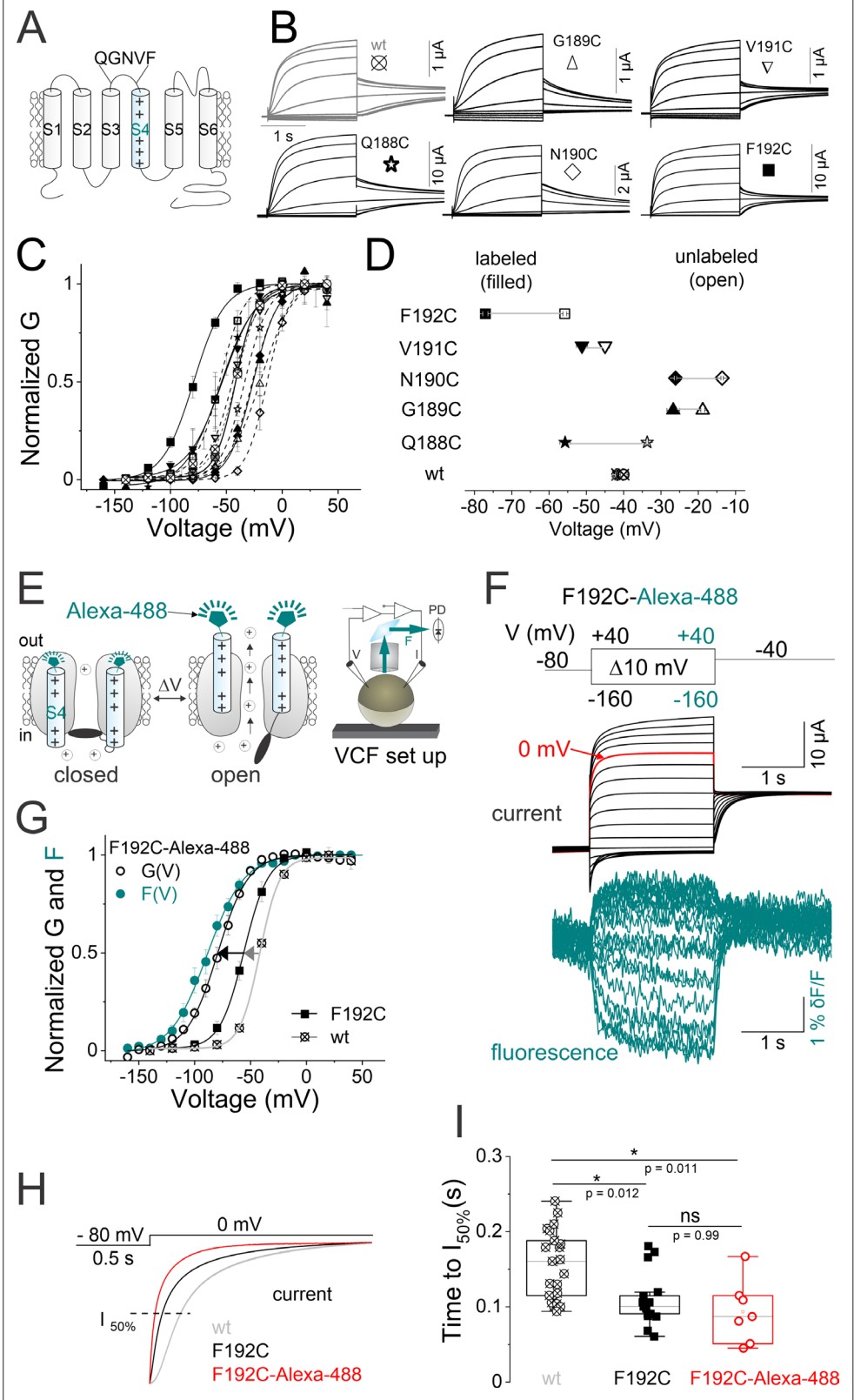

**Figure 2.** Labeled KCNQ2-F192C channels track S4 movement. (**A**) Cartoon showing the topology of one KCNQ2 subunit and the residues in the S3–S4 linker that were sequentially mutated to cysteine. (**B**) Currents from oocytes expressing a series of cysteine mutants in the S3–S4 linker of KCNQ2 channel. Cells are held at −80 mV and stepped to potentials between −140 mV and +40 mV in 20 mV steps for 2 s followed by a tail to −40 mV.

*Figure 2 continued on next page*

*Figure 2 continued*

(**C**) Normalized G(V) (lines from a Boltzmann fit) curves from (open symbols) unlabeled and (filled symbols) Alexa-488-maleimide labeled wt and cysteine mutations shown in (**B**). The midpoints of activation for the fits are shown in **Supplementary file 1**. Data are mean ± SEM, n=5–24; see Materials and methods. (**D**) Summary of $G(V)_{1/2}$ values for the wt and cysteine mutants (open symbols) before and (filled symbols) after Alexa-488-maleimide labeling. (**E**) Cartoon representing the voltage clamp fluorometry (VCF) technique. A cysteine is introduced at position 192 (close to the voltage sensor [S4]) and labeled with a fluorophore tethered to a maleimide group (Alexa-488–5 maleimide). Upon voltage changes, labeled-S4s move and the environment around the tethered fluorophore changes, altering fluorescence intensity. Both current and fluorescence are recorded simultaneously using a VCF set up. (**F**) Representative current (black) and fluorescence (cyan) traces from Alexa-488-labeled KCNQ2-F192C channels (KCNQ2*) for the indicated voltage protocol (top). A sweep to 0 mV is depicted in red to facilitate comparison of time courses in (**H–I**). (**G**) Normalized G(V) (black solid lines from Boltzmann fit) and F(V) (cyan circles and cyan solid line from a Boltzmann fit) curves from (black circles) F192C-Alexa-488 'KCNQ2*', (black squares) unlabeled F192C, and (gray squares) wt channels. The midpoints of activation for the fits are: $GV_{1/2 F192C-Alexa-488}$ = –77.1 ± 2.7 mV, (n=9), $FV_{1/2 F192C-Alexa-488}$ = –87.1 ± 3.9 mV, (n=8), $GV_{1/2 unlabeled-F192C}$ = –55.8 ± 0.8 mV, (n=9), and $GV_{1/2 wt}$ = –43 ± 0.7 mV, (n=21), **Supplementary file 1**. Data are mean ± SEM; see Materials and methods. (**H**) Representative current time courses of (gray) wt, (black) F192C, and (red) F192C-Alexa-488 channels in response to the protocol shown on top. The dashed line represents 50% of the maximum current level at the end of the depolarizing pulse. (**I**) The time courses of current activations are quantified as the time to reach half the maximum current level at the end of the depolarizing pulse in (H, dashed line). Data are presented as mean ± SEM, n=9–21. Statistical significance was determined using ANOVA and Tukey's post hoc test, and significance level was set at p<0.05. Asterisks denote significance: p<0.05*. V: voltage; PD in this cartoon represents: photodiode photodetector.

The online version of this article includes the following figure supplement(s) for figure 2:

**Figure supplement 1.** Cysteine-scan mutagenesis of S3–S4 linker identifies F192C as the ideal position for fluorophore labeling.

*supplement 1C*, right panel). The fluorescence signals in *Figure 2F* (and *Figure 2—figure supplement 1B*) have a non-linear voltage dependence and are much slower than the voltage changes per se, which suggest that the fluorescence changes are not electrochromic responses of the dye to voltage changes. Importantly, the changes in fluorescence signal are likely caused by Alexa488-maleimide (or by Dylight488-maleimide) attached to F192C as oocytes expressing wild-type KCNQ2 channels treated with either fluorophore do not show a voltage-dependent fluorescence signal (*Figure 2—figure supplement 1E,F*). While KCNQ2-F192C channels labeled with both fluorophores render robust fluorescence signals (*Figure 2F* and *Figure 2—figure supplement 1B*), we use Alexa488-maleimide to label KCNQ2-F192C (henceforth called KCNQ2*) throughout the study.

*Figure 2F* shows VCF from labeled KCNQ2* channels in response to a family of voltage steps (from –160 to +60 mV). The steady-state fluorescence/voltage curve (F[V]), tracks the G(V) of KCNQ2* channels ($F(V)_{1/2 F192C}$ = –87.1 ± 3.9 mV, n=8 and $G(V)_{1/2 F192C}$ = –77.1 ± 2.7 mV, n=9, *Figure 2G* and *Supplementary file 1*). The gating properties of KCNQ2* channels (G[V] and the time courses) deviate from that of wt and unlabeled KCNQ2-F192C channels (*Figure 2G–I*). Labeling F192C with Alexa488-maleimide (or with Dylight488-maleimide) shifts the G(V) relationship to negative voltages relative to unlabeled KCNQ2-F192C and wt channels ($\Delta GV_{1/2}$ = –21.3 ± 0.8 mV and $\Delta GV_{1/2}$ = –35.4 ± 2.2 mV, respectively, *Figure 2C, D and G* and *Figure 2—figure supplement 1D*). Moreover, compared to wt KCNQ2 channels, both labeled and unlabeled F192C accelerate the time course of current activation (*Figure 2H–I*).

Next, we measure the time course of fluorescence signals and ionic currents of KCNQ2* channels during both depolarization-induced activation and repolarization-induced deactivation using VCF (*Figure 3*). We use a prepulse of –120 mV to completely close the channel before stepping to the test voltages (*Figure 3A and C*). The fluorescence signal decreases in response to the prepulse to –120 mV (*Figure 3A and C*, arrow), indicating that not all voltage sensors are in their resting position at the –80 mV holding potential. There is a close correlation between the time course of fluorescence signals and ionic currents at all the voltages tested (*Figure 3B and D*). The close correlations in time (*Figure 3*) and voltage dependences (*Figure 2G*) of S4 motion (fluorescence) and activation gate (ionic current) resemble those observed for homologous KCNQ1 (without KCNE1; *Osteen et al., 2010b*) and KCNQ3 channels (*Kim et al., 2017; Barro-Soria, 2019*).

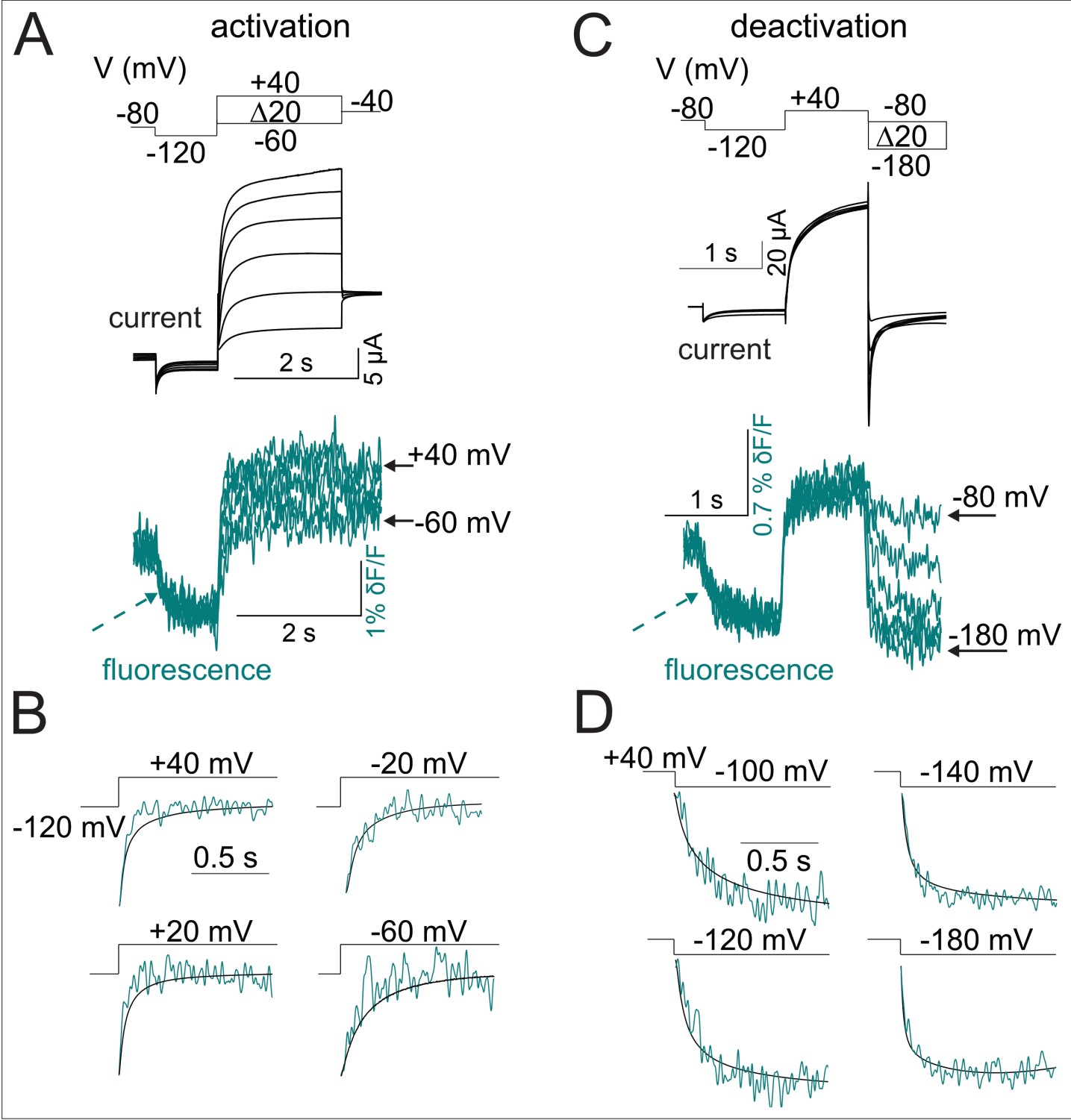

**Figure 3.** Fluorescence from KCNQ2* correlates with channel opening. (**A–D**) Representative current (black) and fluorescence (cyan) traces from KCNQ2* channels for the activation (**A**) and deactivation (**C**) voltage protocols (top). In response to the prepulse to −120 mV, the fluorescence signal decreases (cyan dashed arrow), indicating that not all voltage sensors were in their resting position at the holding potential (−80 mV). Representative experiments showing time courses of (**B**) activation and (**D**) deactivation of current (black) and fluorescence (cyan) signals from KCNQ2* channels at different voltages as in (**A**) and (**C**), respectively. Note that the current and the fluorescence signals correlate during both channel activation and deactivation.

## The voltage dependence of A193C accessibility matches the GV curve of KCNQ2 channels

We use the state dependent modification of A193C by MTSET (*Figure 1—figure supplement 1C*) to measure the rate of access to MTSET at different voltages as an independent assay of S4 movement in KCNQ2 channels (*Figure 4*). External MTSET modification speeds up the activation of A193C channels and increases the current amplitude (*Figure 4B*). While MTSET modifies A193C channels at voltages more positive than −100 mV (*Figure 4B and C*), the rate of MTSET modification of KCNQ2 A193C was fivefold faster at +20 mV compared to −100 mV (*Figure 4C* and *Supplementary file 1*). The modification rate for A193C approaches zero between −140 mV and −160 mV, as if A193C is inaccessible at those voltages (*Figure 4D*). The voltage dependence of the modification rate by MTSET (mod. rate[V]) follows the G(V) for A193C channels (mod. rate[V] = −72.8 ± 24.5 mV, n=3–8 and G[V] = −70 ± 2.4 mV, n=12, *Figure 4D*). Because steady-state conductance/voltage curves, G(V)s, of A193C and labeled F192C channels are similar ($GV_{1/2A193C}$ = −70 ± 2.4 mV, [n=12] and $GV_{1/2F192C-Alexa}$ = −77.1 ± 2.7 mV, [n=9], *Figure 4E and E'*), and under the assumption that these two channels use the same S4 movement to generate these similar G(V)s, we compare the voltage dependence of the modification rate (mod. rate) of A193C with the voltage dependence of the fluorescence of labeled F192C. The mod. rate(V) of A193C has a similar voltage dependence as the F(V), (F[V] = −87.1 ± 3.9 mV, n=8, *Figure 4E*), as if the fluorescence of KCNQ2* accurately reflects S4 movement.

## Disease-causing mutations differentially affect S4 and gate domains

Next, we investigate the mechanism(s) by which the epilepsy-associated mutations R198Q and R214W (*Millichap et al., 2017*; *Castaldo et al., 2002*) alter KCNQ2 channel voltage-dependent activation. R198Q, which neutralizes the first gating charge of S4 in KCNQ2 channels (*Figure 5A*), was previously shown to shift the G(V) to more hyperpolarized potentials and to slow the kinetics of deactivation (*Millichap et al., 2017*). To study the effect of the R198Q mutation, we introduce R198Q into the KCNQ2* background and simultaneously monitor S4 movement (by fluorescence) and gate opening (by ionic current) using VCF (*Figure 5A–C*). In line with a previous report (*Millichap et al., 2017*), we find that compared to KCNQ2* channels, the (labeled) KCNQ2*-R198Q mutation causes a hyperpolarizing shift in the G(V) curve (*Figure 5C*, black arrow and *Figure 5—figure supplement 1E*) and slows the time course of current deactivation (*Figure 5—figure supplement 1B,D*, red). VCF shows that KCNQ2*-R198Q channels exhibit fluorescence signals and ionic currents that continue to closely follow each other in terms of their time courses and voltage dependence of activation (*Figure 5B–B' and C* and *Figure 5—figure supplement 1C,E*, red). Moreover, compared to KCNQ2*, KCNQ2*-R198Q channels exhibit fluorescence signals that are shifted to negative voltages ($\Delta F_{1/2}$ = −32.7 ± 1.4 mV) similar to its negatively shifted G(V) curve ($\Delta G_{1/2}$ = −33.2 ± 1.3 mV) (*Figure 5C* and *Figure 5—figure supplement 1E*). These results suggest that the R198Q mutant, which neutralizes the first gating charge, alters channel function by directly affecting S4 activation.

In contrast to R198Q, the R214W mutation was previously reported to shift the G(V) relationship to more depolarized voltages, to slow the kinetic of current activation, and to accelerate the kinetic of current deactivation (*Castaldo et al., 2002*). We also introduce R214W into the KCNQ2* background and perform VCF (*Figure 5D–E*). Compared to KCNQ2*, KCNQ*-R214W channels display a rightward shifted G(V) curve ($\Delta G_{1/2}$ = +60 mV ± 1.8 mV, *Figure 5E*, black arrow and *Figure 5—figure supplement 1E*), slow the time course of current activation, and accelerate the time course of current deactivation (*Figure 5—figure supplement 1A-D*, maroon), as previously reported for R214W channels (*Castaldo et al., 2002*). VCF shows that in R214W channels, the time course of fluorescence signal precedes the time course of ionic current (*Figure 5D'*, and *Figure 5—figure supplement 1C*, maroon). Interestingly, the F(V) curve of R214W, which is similar to the F(V) curve of KCNQ2* channels, is markedly left shifted compared to its G(V) curve ($FV_{1/2\,R214W}$ = −77 ± 0.6 mV, [n=7] and $GV_{1/2\,214W}$ = −17.1 ± 0.9 mV, [n=8], *Figure 5E* and *Figure 5—figure supplement 1E*). The separation between the F(V) and G(V) curves suggests that R214W dissociates voltage sensor (S4) movement from channel opening. Since R214W is in the loop connecting S4 to the S4–5 linker (not within the voltage sensor itself, *Figure 5—figure supplement 2A*), our data most likely suggests that this mutation affects activation gating without directly affecting the S4 movement which results in fluorescence change.

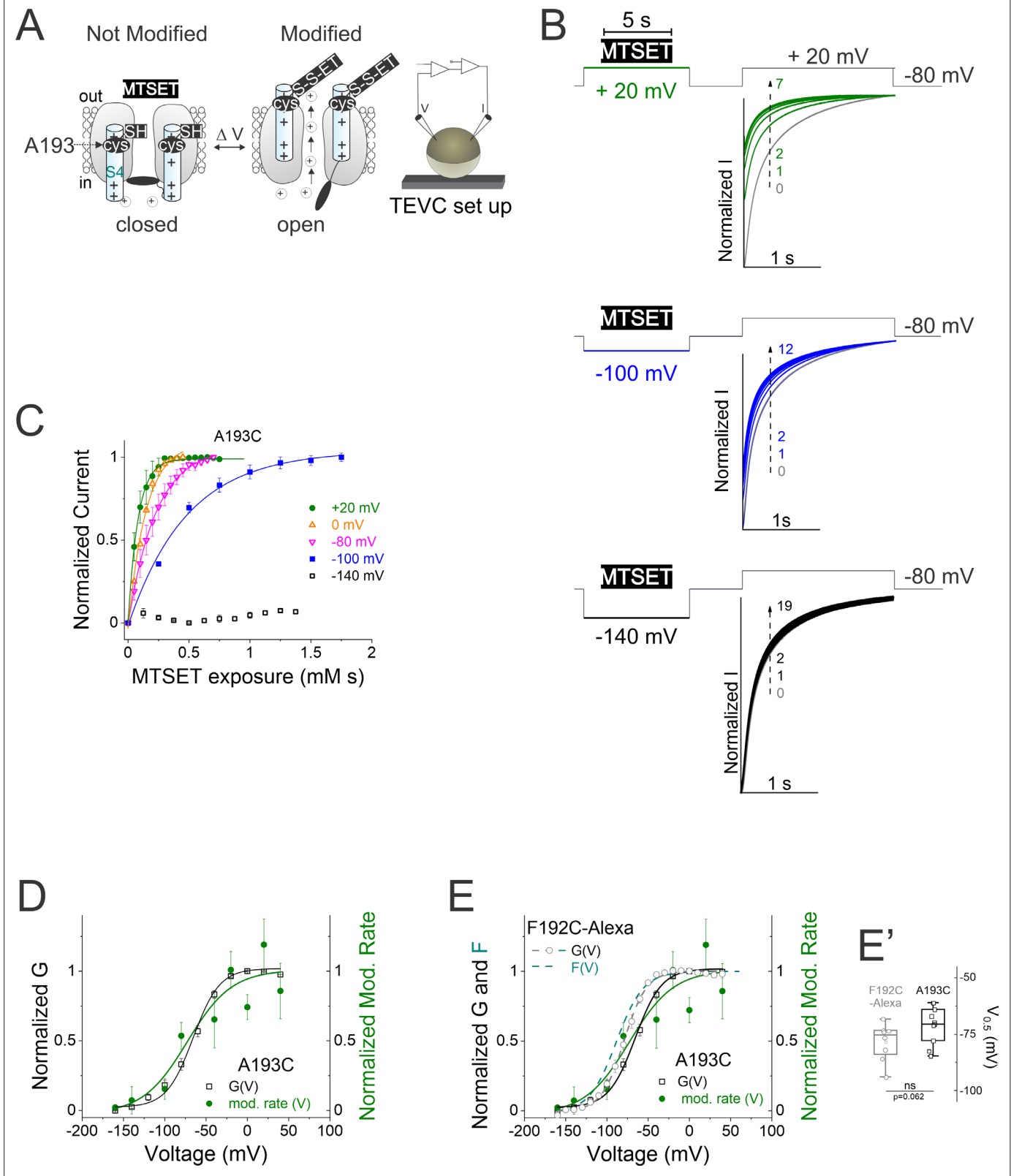

**Figure 4.** Accessibility of residue A193C supports voltage-dependent motion of S4 segment. (**A**) Cartoon representing extracellular cysteine accessibility of residue A193C as in *Figure 1A*. (**B**) Currents in response to +20 mV voltage steps before (gray trace #0) and during several 5 s applications of methanethiosulfonate (MTSET) at +20 mV (green traces #1–7), –100 mV (blue traces #1–12), and –140 mV (black traces #1–19) on A193C channels for the indicated voltage protocol. We repeat MTSET applications (10 µM at +20 and –100 mV, and 20 µM at –140 mV) in between 25 s

*Figure 4 continued on next page*

*Figure 4 continued*

washouts as shown in each voltage protocol. (**C**) Normalized current of A193C during MTSET exposure at +20 mV (green), 0 mV (orange), – 80 mV (pink), –100 mV (blue), and – 140 mV (black). (**D and E**) Normalized G(V) curves (squares and black line from a Boltzmann fit) of A193C channels and voltage dependence of the modification rate (mod. rate [V], green circles and green line from a Boltzmann fit) for MTSET to residue A193C. In (**E**), dashed lines represent 'wt' KCNQ2* (black) G(V) and (cyan) F(V) curves for comparison. (**E'**) Summary of G(V)$_{1/2}$ for (open squares) A193C and (open gray circles) labeled F192C-Alexa channels. Data are presented as mean ± SEM, n=9–12. Statistical significance was determined using paired Student t-test and significance level was set at p<0.05, p=0.062. The midpoints of activation for the fits are: GV$_{1/2A193C}$ = – 70 ± 2.4 mV, (n=12) and GV$_{1/2F192C-Alexa}$ = –77.1 ± 2.7 mV, (n=9); Mod. rate V$_{1/2\ A193C}$ = –72.8 ± 24.5 mV, (n=3–8); GV$_{1/2A193C}$ = – 70 ± 2.4 mV, (n=12); see *Figure 2G* for KCNQ2* GV$_{1/2}$ and FV$_{1/2}$, values.

## Discussion

In this paper, we provide functional data characterizing the voltage sensing mechanism of KCNQ2 channels. We show that during activation, a stretch of S4 residues becomes exposed to the extracellular solution, thereby revealing S4 outward motion. Our fluorescence measurements show a close correspondence between the voltage sensor (S4) movement and channel opening in KCNQ2 channels as both voltage dependence and the time courses of fluorescence and ionic current closely correlate. We find that two epilepsy-associated mutations cause shifts in the voltage dependence of channel opening by two different mechanisms, with the R198Q mutation shifting S4 movement while the R214W mutation uncoupling VSD and channel opening. Our findings shed light on the dynamics and state-dependent molecular rearrangements that lead to channel gating. Since KCNQ2 channels play a pivotal role in controlling neuronal excitability, these results provide critical clues to aid in our understanding of the impact of channelopathies on neuronal function. Understanding how mutations affect channel activity can lead to better ways to correct these mutational defects.

Using a state-dependent cysteine modification approach, we map the extracellular boundaries of S4 residues during membrane depolarization. Our cysteine accessibility data suggests that a stretch of 8–9 amino acids (~193 to 200–201), about half of the 17–19 residues forming the S4(21), moves from a membrane-buried position in the resting state to the extracellular solution during activation gating. Previously, mutagenesis and disulfide crosslinking of substituted cysteines or metal-ion bridge experiments inferred putative closed-resting states of S4 of KCNQ2 channels (*Gourgy-Hacohen et al., 2014*). In this study, the first and second positively charged residues of S4 (R198 and R201) were assumed to interact with the first and second counter-charge residues (E130 and E140) in the S2 segment. This arrangement positioned the gating charge transfer center in S2 (F137) in between R198 and R201 in what was assumed to be a deep closed-resting state of S4 (*Figure 1—figure supplement 5A*). More recently, the cryo-EM structure of KCNQ2 channels (*Li et al., 2021b*; PDB:7CR0) revealed the snapshot of the channel in its activated (S4 up) state and the pore in the closed state. This structure shows that R198 and R201 have moved about three helical turns outward (upward) from F137 into a position close to or within the extracellular space (*Li et al., 2021b*; *Figure 1—figure supplement 5B*). These rearrangements are in line with our cysteine accessibility data in which residues N-terminal to residue F202 become exposed in the activated state of S4 at strong depolarizations (*Figure 1—figure supplement 5D*). Our cysteine accessibility data also provides a snapshot of the resting state of S4 in which residues C-terminal to residue A193 are buried at hyperpolarization (*Figure 1—figure supplement 5C*), as previously predicted (*Li et al., 2021b*).

To date, the direct measurement of gating charge movement during channel gating (gating current measurements) has not been resolved in KCNQ2 channels (nor in KCNQ3). Therefore, the VCF signals of the F192C mutant reported here represent a valuable tool to study voltage sensing in human KCNQ2 channels. Our study reveals, however, that both labeled and unlabeled F192C mutants alter the gating properties of KCNQ2 channels (*Figure 2* and *Figure 2—figure supplement 1*). Our results show that the S3–S4 loop is highly sensitive to both mutations and dye-conjugations, with some mutants (and labeled conjugates) generating large shifts of the G(V) relation and accelerating the time course of current activation compared to wt-channels. While not optimal, this is not surprising because similar G(V) shifts of about ~ –10 to –30 mV upon Alexa-488 maleimide labeling have been previously reported in this region for other Kv channels, including KCNQ1(35), KCNQ3(40), BK (*Savalli et al., 2006*), and Kv1.5 (*Vaid et al., 2008*). We did not explore these effects any further, but it may be possible that tethering the dyes Alexa/Dylight to F192C in KCNQ2 channels could interfere with S4 such that labeled F192C might prevent complete S4 deactivation or could help stabilize the activated (or help destabilize the resting) conformation of the S4, hence shifting the voltage dependence

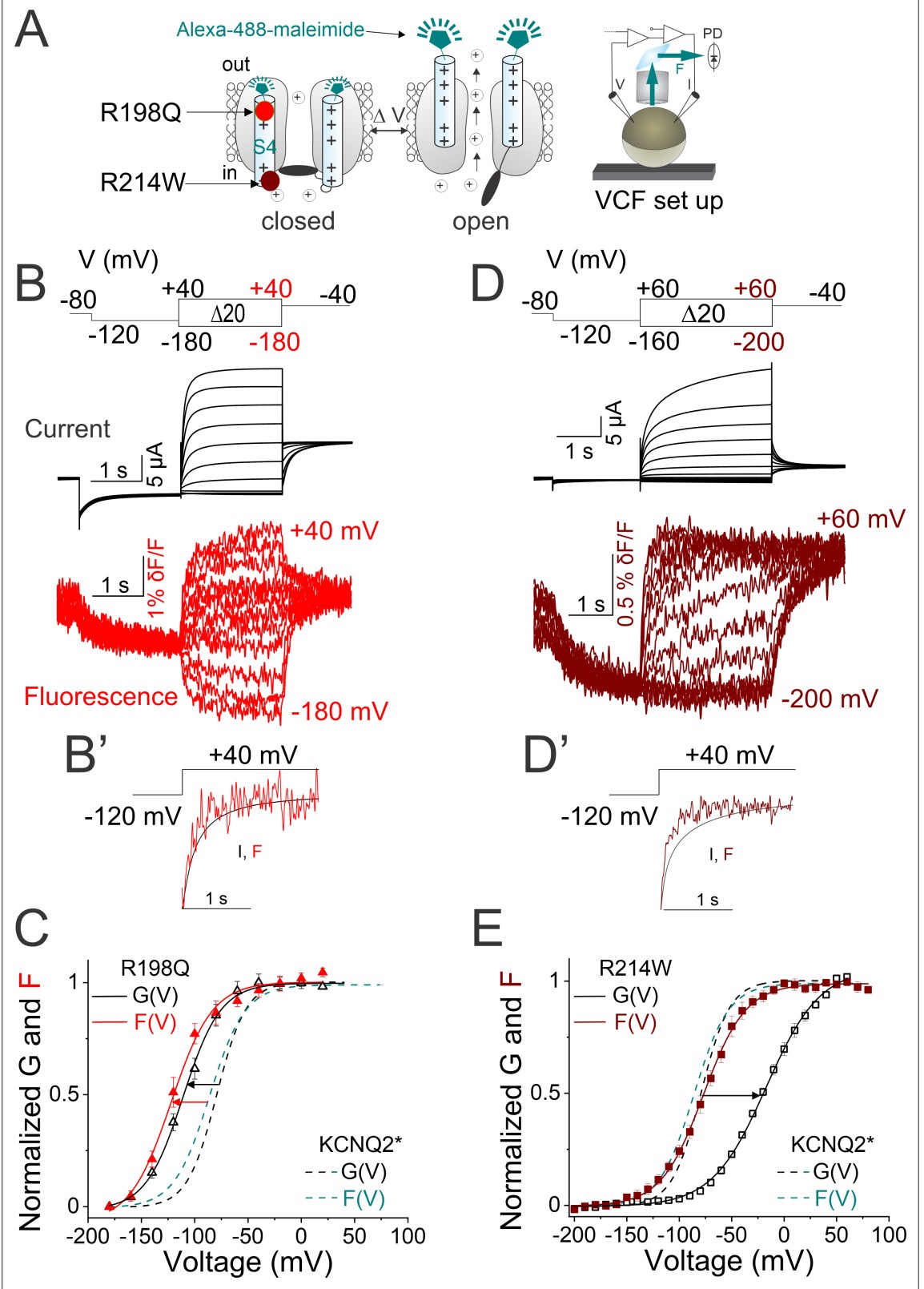

**Figure 5.** Disease-causing mutations in KCNQ2 channels differentially affect S4 and gate domains. (**A**) Cartoon representing the voltage-clamp fluorometry (VCF) technique as in *Figure 2A*. The localization of the two epilepsy-associated mutations - R198Q (red) and R214W (maroon) are shown. (**B**) Representative (black) current and (red) fluorescence traces from KCNQ2*-R198Q channels for the indicated voltage protocol (top). (**B'**) Comparison of the (black) time course of current activation and (red) fluorescence signals from KCNQ2*-R198Q channels in response to the voltage protocol shown.

*Figure 5 continued on next page*

*Figure 5 continued*

(**C**) Normalized G(V) (black triangles and black solid line from a Boltzmann fit) and F(V) (red triangles and red solid line from a Boltzmann fit) curves from KCNQ2*-R198Q. (**D**) Representative (black) current and (maroon) fluorescence traces from KCNQ2*-R214W channels for the indicated voltage protocol (top). (**D'**) Comparison of the (black) time course of current activation and (maroon) fluorescence signals from KCNQ2*- R214W channels in response to the voltage protocol shown. (**E**) Normalized G(V) (black squares and black solid line from a Boltzmann fit) and F(V) (maroon squares and maroon solid line from a Boltzmann fit) curves from KCNQ2*- R214W. (**C and E**) Dashed lines represent KCNQ2-F192C labeled with Alexa-488 (KCNQ2*) G(V) (black) and F(V) (cyan) curves for comparison. The same color code for the two KCNQ2 mutations is shown throughout the figure. The midpoints of activation of the fits are (GV$_{R198Q\ 1/2}$ = −110.3 ± 3.5 mV, (n=10), FV$_{R198Q\ 1/2}$ = −119.8 ± 4.2 mV, [n=4], GV$_{214W\ 1/2}$ = −17.1 ± 0.9 mV, [n=8], and FV$_{R214W\ 1/2}$ = − 77 ± 0.6 mV, [n=7]) and in ***Supplementary file 1***. Data are mean ± SEM.

The online version of this article includes the following figure supplement(s) for figure 5:

**Figure supplement 1.** Gating properties of the epilepsy-associated mutations - R198Q and R214W.

**Figure supplement 2.** PIP2 tightly joints the loop connecting S4 and S4–S5 linker to facilitate channel opening.

to hyperpolarizing voltages and promoting a faster channel opening. These limitations should be taken into considerations in future studies aiming to refine our understanding of the KCNQ gating mechanisms.

Our VCF data shows that both the steady-state voltage dependence and the time course of S4 transitions of fluorescent-labeled KCNQ2 channels closely follow those of the ionic currents, which have virtually no delay and no sigmoidal time course. The close correlations in time and voltage dependence of fluorescence and current of KCNQ2 channels resemble the one-to-one relationship between S4 movement and channel opening reported in KCNQ1 (without KCNE1) (*Osteen et al., 2012*), suggesting that these two homologous channels share similar gating mechanisms. VCF on linked concatemers of KCNQ1 subunits showed that all four voltage sensors can move independently and channel opening can proceed from individual voltage sensor movements (*Osteen et al., 2012*). These findings indicate that S4 does not necessarily require independent conformational changes in all four KCNQ2 subunits before channel opening, as shown for classical voltage-gated K$^+$ channels (*Hodgkin and Huxley, 1990*), but this needs to be further tested in linked-subunit experiments. In KCNQ4 channels, in contrast to KCNQ2, the S4 moves (measured by gating current) much faster than the rate of activation (ionic current), as if the S4 movement was poorly coupled to opening/closing (*Miceli et al., 2012*). Indeed, the gating scheme of KCNQ4 resembles that of the uncoupling R214W mutation (discussed below), whose S4 movement clearly precedes ionic current (*Figure 5D–D'*). Interestingly, the subunit composition of KCNQ channels in the nervous system seems to exhibit higher flexibility and heterogeneity than previously assumed. For instance, besides the well-characterized KCNQ2/3 and KCNQ3/5 heteromeric channels found in the brain, a recent report shows that KCNQ2 not only forms multimeric assemblies with KCNQ5 in vivo, but intriguingly, is able to form part of a more diverse KCNQ2/3/5 heteromeric complex (*Soh et al., 2022*). Thus, insight into the subtle differences in voltage-sensing mechanisms among different KCNQ channel family members is important to understand the different physiological functions that these channels play in the nervous (*Delmas and Brown, 2005*), auditory (*Kubisch et al., 1999*), and cardiac (*Wang et al., 1996*) systems.

Our work provides a framework to understand more in-depth pathophysiological mechanisms of KCNQ2 variants. The R198Q mutation in KCNQ2 channels causes infantile spasms with hypsarrhythmia and encephalopathy associated with severe developmental delay (*Millichap et al., 2017*). Compared to KCNQ2*, KCNQ2*-R198Q channels display left-shifted G(V) and F(V) curves. In addition, KCNQ2*-R198Q channels exhibit fluorescence signals and ionic currents that closely overlap in terms of their time courses and voltage dependence of activation (*Figure 5B–B' and C* and *Figure 5—figure supplement 1C,E*, red). Together, these data suggest that the R198Q mutation alters channel function by directly impacting S4 activation. Conversely, fluorescence data from the epilepsy-associated mutation R214W shows a marked separation between the G(V) and F(V) and a faster fluorescence time course compared to the ionic current time course (*Figure 5* and *Figure 5— figure supplement 1*, brown), suggesting that R214W changes the VSD-PD coupling of KCNQ2. How does R214W dissociate voltage sensor movement from channel opening? Unlike the ILT mutation in the Shaker K$^+$ channels, in which pore opening is dissociated from the first VSD activation but coupled to the second (*Pathak et al., 2005*), KCNQ2*-R214W fluorescence signals do not show a second fluorescence component associated with channel opening. This suggests that the R214W and the ILT mutants decouple VSD-PD through different mechanisms, or alternatively through similar mechanisms

but our labeled F192C is unable to resolve the late component of gating charge movement (fluorescence) associated with pore opening. Moreover, previous studies in the related KCNQ1 channel showed that the F351A mutation separated F(V) from G(V) (*Osteen et al., 2010a*), similar to what is seen with the R214W variant in KCNQ2. Mechanistically, it was postulated that KCNQ1-F351 may couple S4 to pore opening possibly through a physical interaction of F351 with residues within the S4–S5 linker such that point mutations like F351A would alter the VSD-PD interactions, suggesting that F351A eliminates the intermediate open state (*Osteen et al., 2010a*; *Zaydman et al., 2014*; *Taylor et al., 2020*). However, unlike the F351 residue that is localized in the S6 helix (PD) pointing toward the S4–S5 linker of KCNQ1 channels (*Sun and MacKinnon, 2020*), the R214 residue of KCNQ2 lies in the loop region that connects the S4 to the S4–S5 linker (*Li et al., 2021b*; *Figure 5—figure supplement 2A*), as if F351A and R214W may also decouple VSD-PD through different mechanisms.

The recent cryo-EM structures of KCNQ1, KCNQ4, and KCNQ2 channels have provided insights into how mutations in the N-terminal portion of the S4-S5 linker may alter channel gating (*Li et al., 2021b*; *Sun and MacKinnon, 2020*; *Li et al., 2021a*). The cryo-EM structures of KCNQ1 and KCNQ4 show $PIP_2$ bound to these channels close to the S4/S4–S5 interface, in a positively charged pocket. Superimposing the KCNQ2 structure (*Long et al., 2005*) with the homologous structure of KCNQ1 bound to $PIP_{2(55)}$ (*Figure 5—figure supplement 2B*), we noted that residue R214 lies very close to $PIP_2$, suggesting that R214 in the KCNQ2 channel could form part of the positively charged pocket that coordinates $PIP_2$ binding (*Figure 5—figure supplement 2B*). Previous studies in KCNQ1 and KCNQ3 channels have shown that $PIP_2$ directly affects the VSD-PD coupling (*Kim et al., 2017*; *Zaydman et al., 2013*), but the molecular details remain unknown. Based on our VCF results (*Figure 5D and E*), which suggest that the R214W mutant dissociates S4 movement from channel opening, we hypothesize that the positive charge of residue R214 is crucial for $PIP_2$ binding. We propose that in KCNQ2, $PIP_2$ may act like a molecular 'glue' that tightly ties the loop connecting S4 and S4–S5 linker such that during depolarization, the S4 movement effectively pulls S4–S5 away from the pore domain to activate potassium conductance. Therefore, charge-neutralizing mutations like the R214W variant, would affect $PIP_2$ binding and, thereby, weaken the VSD-PD coupling. Supporting this idea, previous studies on KCNQ2 channels bearing the charge neutralizing mutations, R214Q or R214W, found that the loss of the positive charge, and not changes in residue size, was the main functional effect of these disease-associated mutations as both smaller hydrophilic glutamine and bulkier aromatic tryptophan residues at position 214 favored the resting conformation of S4 and, as such, promoted more channel closure (*Miceli et al., 2008*).

One important goal of modern precision medicine is to develop potent/selective therapeutics targeting voltage-gated ion channels. We show that the KCNQ2 variants R198Q and R214W alter the relationship between VSD conformation and gating through different mechanisms. Understanding the impact of human mutations in key regions of the channel, such as the VSD and the pore, will facilitate the prediction of compounds that most effectively restore functionality to specific channel mutations while minimizing potential off-target effects. Small-molecule modulators of KCNQ2 channels have been identified, including the pore opener retigabine (*Maljevic and Lerche, 2014*; *Brodie et al., 2010*; *Wuttke and Lerche, 2006*) and the VSD-targeting ICA family of compounds (*Wulff et al., 2009*; *Wickenden et al., 2008*; *Roeloffs et al., 2008*). Retigabine exhibits poor specificity between KCNQ channel subunits (except for KCNQ1) possibly due to its binding to the S5 segment of the pore (*Wuttke et al., 2005*), which in contrast to the highly diverse VSD region, shows a more conserved sequence among Kv channels. Pore openers like retigabine, which cause a hyperpolarizing shift in the voltage dependence of activation, might seem like a suitable choice to effectively target VSD-PD uncoupling mutations like R214W, but off-target effects on other KCNQ subunits would need to be considered. Unlike retigabine, ICA-like compounds act on the VSD, a less conserved region compared to the pore, presumably allowing ICA to distinguish between KCNQ subunits (*Wickenden et al., 2008*; *Padilla et al., 2009*). Therefore, ICA-like compounds (but in a manner that rightward shifts its voltage dependence) would be more effective to target mutations like R198Q that disturb the VSD. Studies like our, aiming to understand how disease-associated mutations disrupt channel function, will help laying the groundwork for the development of mutation-specific antiepileptic therapies.

In summary, the results presented in this paper provide a foundation to mechanistically understand the voltage-controlled S4 activation that promotes KCNQ2 channel opening. Our cysteine accessibility and fluorescence data add to the existing biophysical and chemical tools to study how KCNQ2

channels open and close the pore in response to changes in the transmembrane voltage. Our findings provide a mechanistic framework to understand how disease-associated mutations may affect channel gating and how drugs can modulate channel function. Understanding which parameters are affected could provide insight into what region may cause channel dysfunction, as exemplified in the epilepsy-associated uncoupling KCNQ2-R214W mutation.

# Materials and methods

## Chemicals

(2-[Trimethylammonium]ethyl)methanethiosulfonate bromide (MTSET) and sodium (2-sulfonatoethyl) methanethiosulfonate (MTSES) were purchased from Toronto Research Chemicals Inc (Downsview, ON, Canada). Alexa Fluor 488 C5-maleimide and Dylight-488-maleimide were purchased from Thermo Fisher Scientific (Waltham, MA, USA). All other chemicals were obtained from Sigma-Aldrich (St. Louis, MO, USA).

## Molecular biology

The full-length human KCNQ2 construct (NCBI Reference Sequence: NP_742105.1; GI: 26051264) was synthesized (GenScript USA, Piscataway, NJ) and ligated between the BamHI and Xbal sites in the multiple cloning sites of the pGEM-HE vector. This vector had been previously modified to contain a T7 promoter and 3' and 5' untranslated regions from the *Xenopus* β-globin gene (*Barro-Soria, 2019*). A BglII restriction site (AGATCT) and a Kozak consensus sequence (GCCACC) were added before the start codon (ATG) of the KCNQ2 gene. Point mutations were made in the KCNQ2 gene using the Quikchange XL site-directed Mutagenesis kit (Agilent) according to the manufacturer's protocol. The correct incorporation of the specific variant was assessed by Sanger sequencing (sequencing by Genewiz LLC, South Plainfield, NJ). The RNA was synthesized in vitro using the mMessage mMachine T7 RNA transcription kit (ThermoFisher Scientific) from the linearized cDNA. mRNA (40–50 nL) was injected into *Xenopus leavis* oocytes (purchased from Ecocyte) using a NanojectII nanoinjector (Drummond Scientific), and electrophysiological experiments were performed 2–5 days after injection.

## Cysteine accessibility measurements in TEVC recordings

We performed cysteine accessibility to MTS reagent (2-[ammonium]ethyl) methanethiosulfonate (MTSET) in two-electrode voltage clamp (TEVC) recordings as previously described (*Larsson et al., 1996*; *Barro-Soria, 2019*). Regular ND96 solution for TEVC contained 96 mM NaCl, 2 mM KCl, 1 mM MgCl$_2$, 1.8 mM CaCl$_2$, and 5 mM HEPES (pH = 7.5). Stock concentrations of 100 mM MTS reagents were prepared in distilled water (prechilled to +4°C) and stored at −20°C until needed. The MTSET was diluted to the appropriate concentration in ND96 solution just prior to each recording (~30 s prior to perfusion) and kept on ice for 30 min maximum. We delivered high K$^+$ solution (100 mM KCl, 1.8 mM CaCl$_2$, 1 mM MgCl$_2$, 5 mM HEPES, pH 7.5, adjusted with KOH) before each day of experiments (prior to application of MTSET) to check that the rate of washin and washout of solutions was fast enough to deliver short durations of MTS- reagents to the oocyte (*Figure 1—figure supplement 2*). A computer-driven, valve controlled, home-made perfusion system that allowed for a rapid switching (within 2 s) between ND96 and MTS reagents during either the open or closed protocol.

We adapted the open and closed state protocols (*Larsson et al., 1996*) to study the solvent exposure of the substituted cysteines in S3–S4 and S4 and test whether these cysteine residues were exposed in open and/or closed channels using irreversible covalent modification by MTSET (*Figure 1B*). Briefly, cells were held at −80 mV for 1 s before stepping to +20 mV for 12 s, then repolarized for another 12 s to −80 mV (for the open state) or voltages between −80 and −140 mV (for the closed state), before stepping to the test potential (+20 mV) to measure the change in currents induced by several 5 s cycles of MTS reagents (see black rectangles in *Figure 1C*, top protocol). We repeat 5 s MTSET application in between 25 s washouts for 8–12 cycles, as shown in the open and closed protocols in *Figure 1C*. MTSET concentrations were between 10 and 100 µM. Ionic currents were recorded in TEVC using an OC-725C oocyte clamp (Warner Instruments), low-pass filtered at 1 kHz and sampled at 5 kHz. Microelectrodes were pulled using borosilicate glass to resistances from 0.3 to 0.5 MΩ when filled with 3 M KCl. Voltage clamp data were digitized at 5 kHz (Axon Digidata 1,440 A; Molecular devices), collected using pClamp 10 (Axon Instruments). The rate of modification was measured by

plotting the change in the current by the MTSET as a function of the exposure to the MTSET (exposure = concentration MTSET [mM] × time [s], measured in [M s]) and fitted with an exponential equation of the form (I[exposure] = $I_0$ exp[−exposure/$\tau$ ]). We then calculated the second-order rate constant from the $\tau$ values (in M s) as $1/\tau = k_{open}$ ($M^{-1}s^{-1}$) of the MTS reaction. Experiments where MTSET modification occurred too quickly (in less than three sweeps) with too high concentrations of MTSET were not included since they cannot accurately be fit with an exponential function to obtain a reliable rate of modification. Results are presented as mean ± SEM (n=number of measurements).

## Voltage clamp fluorometry

VCF experiments were carried out as previously reported (*Barro-Soria, 2019*). Briefly, aliquots of 50 ng of mRNA coding for KCNQ2 or the KCNQ2 variant RNA were injected into *Xenopus laevis* oocytes. At 2–5 days after injection, oocytes were labeled for 30 min with either 100 µM Alexa-488 maleimide or 100 µM DyLight-488 maleimide (Thermo Fisher Scientific) in high ($K^+$) solution (98 mM KCl, 1.8 mM $CaCl_2$, 1 mM $MgCl_2$, 5 mM HEPES, pH 7.05) at 4°C, in the dark. The labeled oocytes were then rinsed three to five times in dye-free ND96 and kept on ice before each recording to prevent internalization of labeled channels. Oocytes were placed into a recording chamber animal pole 'up' in ND96 solution (pH 7.5 with NaOH), and electrical measurements were carried out in TEVC using an Axoclamp 900 A amplifier (Molecular devices). Microelectrodes were pulled to resistances from 0.3 to 0.5 MΩ when filled with 3 M KCl. Voltage clamp data were digitized at 5 kHz (Axon Digidata 1550B via a digital Axoclamp 900 A commander, Molecular devices) and collected using pClamp 10 (Axon Instruments). Fluorescence recordings were performed using an Olympus BX51WI upright microscope. Light was focused on the top of the oocyte through a 20× water immersion objective after being passed through an Oregon green filter cube (41,026; Chroma). Fluorescence signals were focused on a photodiode and amplified with an Axopatch 200B patch clamp amplifier (Axon Instruments). Fluorescence signals were low-pass Bessel-filtered (Frequency devices) at 100–200 Hz, digitized at 1 kHz, and recorded using pClamp 10. When needed, we added 100 µM $LaCl_3$ to the batch solution to block endogenous hyperpolarization-activated currents. At this concentration, $La^{3+}$ did not affect G(V) or F(V) curves from KCNQ2 channels.

## Modeling

The homology model of KCNQ2 channels with S4 in the resting (down) state was created using the Swiss-model program (https://swissmodel.expasy.org/) with the model of KCNQ1 in the resting state (*Kuenze et al., 2019*), as template. All images were created in UCSF ChimeraX, version 1.1 (2020-10-07).

## Electrophysiology data analysis

Data were analyzed with Clampfit 10 (Axon Instruments, Inc, Sunnyvale, CA), OriginPro 2021b (Origin-Labs Northampton, MA), and Corel-DRAW Graphics Suite 2021 software. To determine the ionic conductance established by a given test voltage, a test voltage pulse was followed by a step to the fixed voltage of –40 mV (tail), and current was recorded following the step. To estimate the conductance G(V) activated at the end of the test pulse to voltage V, the current flowing after the hook was exponentially extrapolated to the time of the step and divided by the offset between –40 mV and the reversal potential. The conductance G(V) associated with different test voltages V in a given experiment was fitted by the relation:

$$G(V) = A1 + (A2 − A1)/(1 + \exp(−ze(V − V_{1/2})/k_B T)) \tag{1}$$

where A1 and A2 are conductances that would be approached at extreme negative or positive voltages, respectively, $V_{1/2}$ is the voltage that activates the conductance (A1 + A2)/2, and z is an apparent valency describing the voltage sensitivity of activation (e is the electron charge, $k_B$ is the Boltzmann constant, and T is the absolute temperature). Due to the generally different numbers of expressed channels in different oocytes, we compare normalized conductance, G(V):

$$G(V) = G(V)/A2 \tag{2}$$

Fluorescence signals were corrected for bleaching and time-averaged over 10–40 ms intervals for analysis. The voltage dependence of fluorescence F(V) was analyzed and normalized (F[V]) using relations analogous to those for conductance (equations 1 and 2).

## Statistics

All experiments were repeated four or more times from at least three batches of oocytes. Pairwise comparisons were achieved using paired Student's t-test or one-way ANOVA with a Tukey's test. Data are represented as mean ± SEM, and 'n' represents the number of experiments.

## Acknowledgements

We thank Drs. Derek Dykxhoorn and Hans Peter Larsson for helpful comments on the manuscript. This work was supported by the National Institutes of Health (1R01NS110847) to Rene Barro-Soria.

## Additional information

### Funding

| Funder | Grant reference number | Author |
| --- | --- | --- |
| National Institute of Neurological Disorders and Stroke | R01NS110847 | Rene Barro-Soria |

The funders had no role in study design, data collection and interpretation, or the decision to submit the work for publication.

### Author contributions

Michaela A Edmond, Data curation, Formal analysis, Visualization, Writing – original draft, Writing – review and editing; Andy Hinojo-Perez, Data curation, Formal analysis, Visualization; Xiaoan Wu, Data curation; Marta E Perez Rodriguez, Data curation, Resources; Rene Barro-Soria, Conceptualization, Funding acquisition, Investigation, Methodology, Project administration, Resources, Software, Supervision, Validation, Visualization, Writing – original draft, Writing – review and editing

### Author ORCIDs

Xiaoan Wu http://orcid.org/0000-0003-2098-7298
Rene Barro-Soria http://orcid.org/0000-0003-4804-2739

### Decision letter and Author response

Decision letter https://doi.org/10.7554/eLife.77030.sa1
Author response https://doi.org/10.7554/eLife.77030.sa2

## Additional files

### Supplementary files

• Supplementary file 1. Biophysical properties of wild type and mutant KCNQ2 channels, $V_{1/2}$ and $F_{1/2}$ of activation; $V_{1/2}$ of state dependent MTS modification, and the second-order rate constant of KCNQ2 channels. Data are mean ± SEM, n=number of cells.

• Transparent reporting form

### Data availability

All data generated or analyzed during this study are included in the manuscript and supplementary information (all combined in one pdf file).

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
