## [Editor Report]

This study makes an important technical advance with measurements of voltage-dependent conformational changes of KCNQ2/Kv7.2 channels, measurements which are known to be extremely difficult for this biologically important channel. This advance sheds light on the mechanism by which two human mutations act and opens the door to further investigations of voltage sensor movement in these channels.

---

## [Decision Letter]

**Decision letter after peer review:**

Thank you for submitting your article "Distinctive mechanisms of epilepsy-causing mutants discovered by measuring S4 movement in KCNQ2 channels" for consideration by eLife. Your article has been reviewed by 3 peer reviewers, including Jon Sack as the Reviewing Editor and Reviewer #1, and the evaluation has been overseen by Kenton Swartz as the Senior Editor.

The reviewers and editors appreciate the importance of establishing KCNQ2 VCF, the insights it provides into mechanisms of gating, and how this could enable future mechanistic studies. We think such an advance could be appreciated by the readers of eLife. While we are sorry to say the manuscript will not be accepted in current form, it seems possible that substantial revisions might improve the manuscript. If all of the reviewer concerns can be effectively addressed we would be willing to review a revised manuscript. Here we highlight the types of reviewer concerns (details in reviewer comments) that would be the most essential revisions:

1) Improve or remove the kinetic model. The model presented in Figure 6 is insufficiently described and seems to have several flaws including violation of microscopic reversibility. It is not clear if the model effectively builds on or supports the experimental findings. If a model is to be presented, please give a clear description and rationale for its structure and all the choice of all parameters.

2) Improve reporting of MTS modification and interpretation, especially that of N190C.

3) Generally tighten logic of interpretation. The reviewers point out a number of instances where the narrative of the manuscript has an unclear relation to the data presented and data seem over interpreted. Especially concerning were arguments about the number of voltage sensors moving during gating.

In summary, we appreciate the technical achievement of establishing KCNQ2 VCF, and hope for a manuscript where the evidence for every claim is clearly communicated, with limitations, caveats, and alternate interpretations shared as well.

*Reviewer #1 (Recommendations for the authors):*

I applaud the authors on their thorough characterization of KCNQ2 voltage sensor movements. the distinctions between KCNQ2 and KCNQ1 seem really intriguing. As do the investigations of the gating modulation by R214W.

Suggestions of where the science and its presentation might be strengthened:

A) Address the possibility the only a subset of voltage sensor movements are reported by the fluorophores at position 192. For example, is seems possible that early, independent movements of voltage sensors are totally missed by the fluorescence.

B) Improve the description of how the Markov-chain model is constructed, and the logic of its parameterization. The model seemingly has the form of an MWC gating model, but it is modified strangely and under described. The results describe the gamma and delta transitions as voltage sensor movements and in the model these are combined with a channel opening parameters (L, f) controlling the opening rate. This all seems strange and not well explained. How much of each voltage sensor movement component contributes to fluorescence is also unclear, and hard to make sense of.

C) Make it clearer what data supports each claim in the results. For example, claims are made repeatedly about kinetics while showing only exemplar data and irreversible changes are mentioned but not backed up. There were some seeming discrepancies between the Results narrative and the data (specific instances described below).

D) Further discussion of how understanding the impacts of human mutations on voltage sensor vs pore movements could be valuable. Perhaps this could be in the context of KCNQ drugs that act on the pore, like retigabine, vs other that act on the voltage sensor.

Specific suggestions:

Page 6 "Modification of N190C channels quickly and irreversibly increases the current amplitude, speeds up the kinetic of activation, and changes the voltage dependence of activation in either closed or open N190C channels, albeit to distinct V1/2 values (as also shown earlier for HCN1 channels (40), Figure 1D, E), as if N190C is always accessible and exposed to the extracellular solution (Figure 1K, yellow). "

Quantitation of quickly, irreversibly, current amplitude, kinetic of activation, seem to be lacking, aside from showing exemplar traces. For N190C, evidence of changes in the voltage dependence of activation at -80 mV seem to be lacking: slightly different V1/2 values are in Supplement Table 1, but differences aren't compelling in Fig 1E. Unclear why the -80 mV MTSET differences in V1/2 are considered changes in the voltage dependence of activation for N190C, but not R198C for example. More discussion would be helpful of how modification of N190C at -80 mV could produce a different result than +20 mV, yet still be similarly accessible. It would seem helpful to quantitate the changes in current amplitude in response to the MTSET, for example in the case of N190C at -80 mV and elsewhere that the change in current amplitude is the evidence of modification.

Page 8 "Compared to unlabeled KCNQ2* channels, the time courses of ionic currents of labeled KCNQ2* (labeled with either fluorophore) are similar"

Suggest providing quantitative backing for this claim. The time courses look faster with fluorophore.

Figure 3C Could the Tau on from fluorescence with 20 ms time constants be limited by the 100-200 Hz filtering of the optical signal? Error bars seem to be missing.

Figure 3F The -180 mV fit appears to have a decaying component with a negative amplitude?

Page 8 "signals follow a double exponential time course" debatable, as the fits aren't amazing. Might be better to state "appeared to have multiple exponential components and were fit by a double exponential".

Page 9 "These data also suggests that an individual voltage sensor movement might be sufficient to open the channel." State more clearly what data is suggestive of this?

Fig 5B,D Please more quantitatively analyze the reported similarity and difference in F and I kinetics

Page 10 "the ILT mutation in the Shaker K+ channels "

the ILT mutations dissociate early voltage sensor movement from pore opening, but the ILT pore opening remains coupling to late voltage sensor movements and is detected by extracellular fluorescence measurements similar to those employed in this manuscript (doi: 10.1085/jgp.200409197). The fact that a fluorescence component is not observed with pore opening of KCNQ2*-R214W could suggests a different mechanism of decoupling than ILT, or that the late component of gating charge movement associated with pore opening is not reported by the fluor.

Fig 6B Something seems wrong here: the fit is purported to represent 1/alpha, but with zalpha = 0.43 the tau fast at +40 should be 2.7x faster than -20 mV, and this is not the case.

Fig 6C Tau on F slow as well as Tau on G slow (Fig 3 C) seem to lose their voltage dependence at more positive voltages. This could mean pore opening itself (gamma) has little voltage dependence.

Fig 6F Why connote that f is always to the 1st power? Do L and f only impact the opening rate? The modeling is not described sufficiently to reproduce it. alpha and beta are missing altogether. zdelta/zgamma should also be listed in Supplement Table 2

Page 11 "We find no experimental evidence supporting a constitutive open state (O0 in Figure 6F, gray)" From the model you can calculate the expected Popen at very negative voltages, it could be just a very low Popen.

Page 12 "By decreasing the opening transition (L in Figure 6F) relative to wt KCNQ2 channels, the model also describes the clear separation between F(V) and G(V) curves observed in mutated KCNQ2*-R214W channels (Figure 6H), under the assumption that R214W changes the voltage sensing domain-pore domain (VSD-PD) coupling such that it prevents opening before multiple S4 have activated (Figure 6F, dashed maroon arrow)."

Difficult to parse this sentence. In addition to L, f and gamma are also changed. What changes VSD-PD coupling, isn't that L?

Page 12 "Additionally, data from the R198Q mutation can also be simulated by shifting the voltage dependence to negative voltages. " I imagine this is probably right but the claim is not justified by simulations.

Page 13 Confidence in estimated total gating charge of 7.37 e0 per channel moved during KCNQ2 activation gating is limited due to the issue poorly described parameterization of the gating model.

Page 14 "the overall voltage-dependent gating mechanisms of KCNQ2 is qualitatively similar to that of KCNQ1". To me it seems that the overall voltage-dependent gating mechanism of KCNQ2 is qualitatively distinct from KCNQ1.

*Reviewer #2 (Recommendations for the authors):*

1. In Abstract and other places, the sentences such as "channel opening does not require multiple VSD movements" are not clear. Do the authors try to say, "the movements of multiple VSDs" or "the movements of VSDs in multiple steps"?

2. Did N190C really modify the channel at both -80 mV and +20 mV? Why would the same covalent modification of the channel at two voltages result in different GV relations (Fig 1E)? Is there another Cys in the channel that was modified differently at the two voltages? Did the authors use different protocols to measure GV in these two conditions? These need to be explained. The authors claimed a similar result in a reference, but it was not obvious if the reference showed the similar result or explained the result. The authors can cite the reasons given by the reference (40) if these can make sense of the results in Fig 1E.

3. Fig 1J at -80 mV: why is the current amplitude 0? It does not seem to be consistent with the description in the legend or J'.

4. Fig 2C and Fig 2-FigS1E: The FV with Alexa labeling increases at voltages >0 mV. Is this real or artifact? If real, does it indicate a second VSD movement?

5. In Fig 4E, the comparison between MTS modification rate of A193C and FV of KCNQ2* makes no sense. These two curves derived from different mutations and modifications may overlap by coincidence.

6. The data in Fig 6B-E seem to differ from the data in Fig 3C,D although the legend of Fig 6 claims that these are the same data. For instance, in Fig 3C the fast component of F does not show a voltage dependence, but in Fig 6B and C the fast and slow components of F show a similar voltage dependence.

7. The model in Fig 6F raises several concerns: (1) Why do the vertical transitions have the rates of VSD activation, while they should represent pore opening/closing? (2) What does f represent in the scheme? Can it be part of L? (3) Detailed balance is violated in the left-most loop connecting C0, O0, O1, and C1.

8. In Fig 6G, which states were used to represent currents? Which states represent fluorescence? Particularly, with both the horizontal and vertical transitions represent VSD activation, the rationale for simulating fluorescence need to be justified and the methods clearly described.

*Reviewer #3 (Recommendations for the authors):*

I was very happy to read this paper and feel that this work is an important step forward for those working on these channels. A lot of progress has been made with KCNQ1 because of the relative ease of recording VCF signals, whereas similar work in KCNQ2-5 has been difficult. The KCNQ2-5 channels differ significantly from KCNQ1 in terms of their function and auxiliary protein regulation, so the development of useful tools to carry out detailed biophysical studies on these channels is valuable and took quite a heroic effort.

I have quite a few comments, just important things that I would add to the paper in the interest of being thorough and not over-interpreting some of the findings.

1. Page 3. "muscarine-regulated M-current". I would hesitate to call it 'muscarine regulated' as it can be sensitive to a variety of neurotransmitters that signal via Gq (ie. acetylcholine... I agree with the historical perspective of naming the current, but the wording may imply physiological regulation to some readers).

2. Page 9: "These data also suggest that an individual voltage sensor movement might be sufficient to open the channel". The basis for this interpretation is not clear (at least not at this stage of the paper). Also, as mentioned in the public review, a counterpoint to this observation is that the KCNQ2 or KCNQ3 currents typically exhibit a sigmoidal time course (also see Figure 1) to activation which might be accounted for by a requirement for multiple subunits to reach an activated conformation. Could this arise because arise because the dye labeling may prevent complete VSD deactivation or interfere with gating in some other way. This is also brought up at the top of page 14 and I have concerns that this could be a contentious statement. I would suggest more caution when describing and interpreting these properties.

3. One way to potentially address this explicitly (ie. point #2) would be to include a direct comparison of unmodified and modifier I192C (and maybe WT KCNQ2 as well) in Figure 2. It is 100% fine with me that there are differences, but it should be clearly shown and described as a consideration when interpreting data, and some comparison like that would help readers.

4. The other technical concern that I had was about fitting the fluorescence traces and perhaps adding complexity where it is not needed and not necessarily supported by data (perhaps this is being done due to analogy to prior work in KCNQ1). Based on the sample sweeps, there does not usually seem to be a great reason to fit with 2 components (eg. Figure 3) - is it really necessary in this case (ie. would it make a difference in terms of the predicted currents, especially given the uncertainty about sigmoid character of current activation)? A few other issues with the description of the model are that some parameters appear to be missing from Supplemental Figure 2 (ie. Alpha and Beta rates, and z for gamma and delta rates). In the text it seems that the gamma and delta rates are meant to be associated with channel opening, but the large amplitude fast component (alpha+beta rates) seem to correlate with the early stages of channel opening, it seems. Perhaps clarifying this by showing individual fit components, or simplifying the fitting/model would be helpful.

---

## [Author Response]

Essential revisions:1) Improve or remove the kinetic model. The model presented in Figure 6 is insufficiently described and seems to have several flaws including violation of microscopic reversibility. It is not clear if the model effectively builds on or supports the experimental findings. If a model is to be presented, please give a clear description and rationale for its structure and all the choice of all parameters.2) Improve reporting of MTS modification and interpretation, especially that of N190C.3) Generally tighten logic of interpretation. The reviewers point out a number of instances where the narrative of the manuscript has an unclear relation to the data presented and data seem over interpreted. Especially concerning were arguments about the number of voltage sensors moving during gating.

We thank the editor for their helpful comments. In response, we have removed the kinetic model as suggested by all three reviewers. We agree that the model presented in the original Figure 6 was underdeveloped and would need more experimental data to better describe KCNQ2 channel gating. This kinetic model is deleted in the revised version. (2) We have also added new data to improve our understanding and interpretation of MTSET modification data, including MTSET modification of the N190C mutant in both closed and open states, addressing the comments of Reviewer 1 and 2. (3) We tightened our conclusions to the experimental findings by thoroughly and clearly communicating limitations, caveats, and alternative interpretations, particularly to those concerning labeled and unlabeled F192C and restraining from informing about the number of S4 moving during gating in the results, as suggested by reviewers. Specifics on all these points are discussed below in greater detail.

Reviewer #1 (Recommendations for the authors):I applaud the authors on their thorough characterization of KCNQ2 voltage sensor movements. the distinctions between KCNQ2 and KCNQ1 seem really intriguing. As do the investigations of the gating modulation by R214W.Suggestions of where the science and its presentation might be strengthened:A) Address the possibility the only a subset of voltage sensor movements are reported by the fluorophores at position 192. For example, is seems possible that early, independent movements of voltage sensors are totally missed by the fluorescence.

We thank the reviewer for this observation. We agree with the reviewer that early, independent movements of voltage sensors might have been missed by the fluorescence. We have now re-analyzed the data and determined that while the time course of the fluorescence appeared to have multiple exponentials, our fluorescence data lacked sufficient resolution to reliably estimate in detail the first (fast) component. This might be because of the low signal-to-noise ratio of our VCF or/and as correctly noted by reviewer# 1 below, because the filtering might have limited the tau-on from the optical signal (shown to be 20 ms in Figure 3C of the original submission).

That is why, as suggested by reviewers # 3, we have removed the kinetics comparison of fluorescence and current from the revised version of Figure 3 and do not claim the existence of fast and slow fluorescence components. This comparison on the original submission primarily served to support the Markov kinetic model, which has been removed from the revised manuscript. We now simply state: …” There is a close correlation between the time course of fluorescence signals and ionic currents at all the voltages tested (Figure 3B, D). The close correlations in time (Figure 3) and voltage dependences (Figure 2G) of S4 motion (fluorescence) and activation gate (ionic current) resemble those observed for homologous KCNQ1 (without KCNE1)(42) and KCNQ3 channels(41, 43).”

B) Improve the description of how the Markov-chain model is constructed, and the logic of its parameterization. The model seemingly has the form of an MWC gating model, but it is modified strangely and under described. The results describe the gamma and delta transitions as voltage sensor movements and in the model these are combined with a channel opening parameters (L, f) controlling the opening rate. This all seems strange and not well explained. How much of each voltage sensor movement component contributes to fluorescence is also unclear, and hard to make sense of.

We have removed the kinetic model as suggested by all three reviewers. We apologize for the flaws shown in the old Figure 6F regarding the violation of reversibility and for the poor descriptions of its logic and parametrization. We agree that gathering more fluorescence and current data with different protocols to extract all the parameters are needed to better describe KCNQ2 gating.

C) Make it clearer what data supports each claim in the results. For example, claims are made repeatedly about kinetics while showing only exemplar data and irreversible changes are mentioned but not backed up. There were some seeming discrepancies between the Results narrative and the data (specific instances described below).

Thanks! In the revised version, we have backed up exemplar data with appropriate statistical analysis (please see statistics in new Figure 1-figures supplement 1, 3, and 4; Figure 2, Figure 2-figure supplement 1; and Figure 5-figure supplement 1). We have been careful to avoid any unclear/imprecise text and made sure to more closely relate the claims being made to specific experimental observations (Please, see below response with specific suggestions).

D) Further discussion of how understanding the impacts of human mutations on voltage sensor vs pore movements could be valuable. Perhaps this could be in the context of KCNQ drugs that act on the pore, like retigabine, vs other that act on the voltage sensor.

We thank the reviewer for this suggestion. We have provided more context to the proposed distinctive gating mechanisms of mutations affecting either voltage sensors or pore movement. We have now expanded on this idea on pages 15 and 16, last and first paragraphs, respectively of the discussion

Specific suggestions:Page 6 "Modification of N190C channels quickly and irreversibly increases the current amplitude, speeds up the kinetic of activation, and changes the voltage dependence of activation in either closed or open N190C channels, albeit to distinct V1/2 values (as also shown earlier for HCN1 channels (40), Figure 1D, E), as if N190C is always accessible and exposed to the extracellular solution (Figure 1K, yellow). "Quantitation of quickly, irreversibly, current amplitude, kinetic of activation, seem to be lacking, aside from showing exemplar traces. For N190C, evidence of changes in the voltage dependence of activation at -80 mV seem to be lacking: slightly different V1/2 values are in Supplement Table 1, but differences aren't compelling in Fig 1E. Unclear why the -80 mV MTSET differences in V1/2 are considered changes in the voltage dependence of activation for N190C, but not R198C for example. More discussion would be helpful of how modification of N190C at -80 mV could produce a different result than +20 mV, yet still be similarly accessible. It would seem helpful to quantitate the changes in current amplitude in response to the MTSET, for example in the case of N190C at -80 mV and elsewhere that the change in current amplitude is the evidence of modification.

In the first submission, the term “irreversible” was used to describe those MTSET-modified currents measured after extensive washout of MTSET from the bath (referring to permanently modified channels). We apologize for any confusion caused by this statement. In the revised version we have removed imprecise wording like quickly and irreversible. We have also performed additional experiments and the requested analysis of current amplitude and G(V) shifts from wt and the cysteine substitutions at – 80 mV and + 20 mV. This new data is presented in the new Figure 1 and new Figure 1- figure supplements 1, 3 and 4 of the revised manuscript.

In the first submission, we claimed that MTSET modified N190C channels as the current amplitude in both closed and open states increased after MTSET application (old Figure 1D). We also used the MTSETmediated G(V)1/2 shifts in the open and closed states to support that claim. However, as correctly noted by the reviewer, these G(V)1/2 values shown in the original Table 1 (and depicted in the original Figure 1E as G(V) curves) appeared to show different MTSETmediated shifts in the G(V) relationship for closed and open sates. These G(V)1/2 differences raised the concern that N190C might not be accessible in the closed state.

To address the reviewer’s concern about the extent in MTSET-mediated modification of N190C, we repeated the cysteine accessibility experiments in both the closed and open states. We use the same protocol as shown in the original submission to increase the number of trials (which in the original version was n = 3, highlighted in red in the summary table, Author response image 1, provided for the reviewers but not included in the revised manuscript). While this new set of data shows the same trend (more negative MTSET-mediated G(V)1/2 shift in the open state than in the closed state) as also seen in the original data (in red), the extent of this change shows no statistical significance between both states (Author response image 1)

**Author response image 1. sa2fig1:** (a) Currents from oocytes expressing KCNQ2-N190C channels in response to 20-mV voltage steps from – 140-mV to + 40-mV (left) before and (right) after application of MTSET in the open state. (middle) currents in response to a + 20-mV voltage steps during MTSET application on N190C channels in the open state for the indicated voltage protocol. MTSET is applied at + 20-mV for 5-s in between 25-s washouts for 8-15 cycles and the change in current is measured at + 20-mV. (b) Normalized steady-state conductance/voltage relationships, G(V), (lines from a Boltzmann fit) of N190C channels (black) before and (gray ‘closed state’ and yellow ‘open state’) after MTSET application. Summary of (c) relative change in current amplitude and (d) voltage dependence shift of MTSET-mediated modification of N190C channels in the open state. (e) Summary table of the G(V) values from each experiment. Values from the original submission are shown in red. (e’) Statistical analysis from red values. Significance was determined using the paired Student’s t-test and significance level was set at P < 0.05. Asterisks denote significance: p < 0.01**.

However, based on the reviewer’s concern, we adopted a new protocol that more rigorously tested whether MTSET modifies (and modifies to completion) N190C in the closed state. This protocol contains the following modifications (performed for each cysteine mutants in ‘new Figure1 of the manuscript’, but only shown here for N190C for the benefit of the reviewers):

We measured a family of currents in response to 20-mV voltage steps from –140 mV to +40 mV before (Author response image 2) and after applications of MTSET in the closed (Author response image 2’) and open (Author response image 2’) states. To assess the state-dependent modification of substituted cysteines, we first applied MTSET at – 80-mV for 5 s in between 25-s washouts for 8-15 cycles and assayed the change in current at + 20-mV (Author response image 2). On the same cell and after MTSET is washed out of the bath, we repeated a similar protocol but instead applied MTSET at + 20-mV and again assayed the change in current at + 20-mV (Author response image 2). We reason that if MTSET only modifies N190C in the open state, but not in the closed state as it was suggested by the reviewer, the second application of MTSET at depolarized (open) voltages will cause an additional increase in the current amplitude and will shift the G(V) relationship to more negative voltages.

External application of MTSET in the closed state increased the current amplitude and left-shifts the G(V) relationship of N190C channels (ΔGV1/2 N190C closed = – 6.3 ± 1.3 mV, n = 11, Author response image 2’, d-f, gray). We found that after the second MTSET application (now using the open state protocol), there was no additional increase in the current amplitude and the G(V) relationship did not shift further (ΔGV1/2 N190C open = – 7.0 ± 1.7 mV, n = 9, Author response image 1’, d-f, yellow), as would be expected if all N190C channels were fully modified in the closed state. These results (increased in current amplitude and G(V) shift) strongly support our previous conclusion that N190C was accessible in the closed state (S4 in the resting state).

**Author response image 2. sa2fig2:** (a, b’, and c’) Currents from oocytes expressing KCNQ2-N190C channels in response to 20-mV voltage steps from – 140-mV to + 40-mV (a) before and after application of MTSET in the (b’) closed and (c’) open states. (b and c) currents in response to a + 20-mV voltage step during MTSET application on N190C channels in the (b) closed and (c) open states for the indicated voltage protocols. MTSET is first applied at (b) – 80 mV for 5-s in between 25-s washouts for 8-15 cycles and the change in current is measured at + 20-mV. On the same cell and after MTSET is washed out of the bath, MTSET is re-applied at (c) + 20-mV using a similar protocol as in (b). (d) Normalized steady-state conductance/voltage relationships, G(V), (lines from a Boltzmann fit) of N190C channels normalized to peak conductance before MTSET application (black). The G(V) relationships of N190C channels before and after MTSET application in the closed (-80-mV, gray) and open (+ 20-mV, yellow) states are obtained from recordings of panels (a), (b’), and (c’), respectively. Summary of (e) relative change in current amplitude and (f) voltage dependence shift of MTSET-mediated modification of N190C channels in the (gray) closed and (yellow) open states. Mean ± SEM, *n=9-24*. Statistical significance was determined using the Student’s t-test and significance level was set at P < 0.05. Asterisks denote significance: p < 0.001***.

To test whether N190C is also accessible in the open state, we performed a separate experiment in which MTSET is applied at + 20-mV (open) and the change in current is measured at + 20-mV (Figure 1-figure supplement 4). Using this protocol, we found that MTSET also increased the current amplitude and shifted the G(V) relationship of N190C channels (ΔGV1/2 N190C open = – 12.2 ± 10 mV, n = 5, Figure 1-figure supplement 4). Together, these results suggest that N190 was indeed always accessible and exposed to the extracellular solution in both the closed and open states.

Page 8 "Compared to unlabeled KCNQ2* channels, the time courses of ionic currents of labeled KCNQ2* (labeled with either fluorophore) are similar"Suggest providing quantitative backing for this claim. The time courses look faster with fluorophore.

We thank reviewer #1 for this helpful suggestion. Our statement was wrong. In the revised version, we have rewritten this subsection almost in its entirety and show statistics on new panels in Figure 2 and Figure 2-figure supplement 1. We have now added a comprehensive comparison of the time courses between wt, unlabeled KCNQ2-F192C, and labeled-KCNQ2-F192C channels in the new (Figure 2 and Figure 2-figure supplement 1C, and referenced appropriately in the text on pages 8-9 of the revised manuscript).

We now state: “The gating properties of KCNQ2* channels (G(V) and kinetics) deviate from that of wt and unlabeled KCNQ2-F192C channels (Figure 2G-I). Labeling F192C with Alexa488-maleimide (or with Dylight488-maleimide) shifts the G(V) relationship to negative voltages relative to unlabeled KCNQ2-F192C and wt channels (ΔGV1/2 = – 21.3 ± 0.8 mV and ΔGV1/2 = – 35.4 ± 2.2 mV, respectively, Figure 2C, D, G and Figure 2-figure supplement 1D). Moreover, compared to wt KCNQ2 channels, both labeled and unlabeled F192C accelerate the time course of current activation (Figure 2H-I).”

As also suggested for Reviewer# 3, we also show in the revised version a comparison of the time courses of current activation for wt and all scanned cysteine mutants in the extracellular S3-S4 linker (Figure 2-figure supplement 1A and Page 8 of the revised manuscript).

Figure 3C Could the Tau on from fluorescence with 20 ms time constants be limited by the 100-200 Hz filtering of the optical signal? Error bars seem to be missing.

Thanks! Very good point. It might be possible that this is the case, and we acknowledged this technical limitation. We have now re-analyzed the data and concluded that while the time course of the fluorescence appeared to have multiple exponentials, our fluorescence data lacked sufficient resolution to reliably estimate the first (fast) component. This might be because of the low signal-tonoise ratio of our VCF or/and as pointed by the reviewer, because the filtering might have limited the tau-on from the optical signal (shown to be 20 ms in Figure 3C of the original submission).

That is why, as suggested by reviewers # 3, we have removed the kinetics comparison of fluorescence and current from the revised version of Figure 3. This comparison on the original submission primarily served to support the Markov kinetic model, which has been removed from the revised manuscript. We now simply state:

…” There is a close correlation between the time course of fluorescence signals and ionic currents at all the voltages tested (Figure 3B, D). The close correlations in time (Figure 3) and voltage dependences (Figure 2G) of S4 motion (fluorescence) and activation gate (ionic current) resemble those observed for homologous KCNQ1 (without KCNE1)(42) and KCNQ3 channels(41, 43).”

In the original submission, error bars were shown but these were small and difficult to see since the graph was plotted on a log scale. In any case, this analysis has been removed from the revised version.

Page 8 "signals follow a double exponential time course" debatable, as the fits aren't amazing. Might be better to state "appeared to have multiple exponential components and were fit by a double exponential".

Thanks, we agree! Please see response above.

Figure 3F The -180 mV fit appears to have a decaying component with a negative amplitude?

We do not see a decaying component in the other traces we examined at – 180mV. This might be a clamping artifact at this extremely negative voltage. It is very difficult to hold the cell at voltages more negative than – 140 mV for longer than 1 second. In this case where we showed Fluorescence changes at – 180 mV, it is possible that by the end of the pulse some artifacts produced this apparently decaying component.

Page 9 "These data also suggests that an individual voltage sensor movement might be sufficient to open the channel." State more clearly what data is suggestive of this?

We apologize for the inaccuracy of the claim that an individual voltage sensor movement might be sufficient to open the KCNQ2 channel based on the VCF data provided in Figures 2 and 3. We have clarified this in the revised version of the manuscript and this sentence (and its implications) has been removed from the Abstract and Results sections. We only discussed that these alternatives, concerted or independent S4 movement, might equally well explain our VCF data which shows that both the steady-state voltage dependence of S4 transitions and the kinetics closely follow those of ionic currents.

Fig 5B,D Please more quantitatively analyze the reported similarity and difference in F and I kinetics

We thank the reviewer for this helpful comment. We have performed the requested analysis, added new data, and presented the results in Figure 5-figure supplement 1 (which now contains statistical analysis in panels C, D, and E), and, accordingly, amended the text: see Results, subsection “Disease-causing mutations differentially affect S4 and gate domains”

Page 10 "the ILT mutation in the Shaker K+ channels "the ILT mutations dissociate early voltage sensor movement from pore opening, but the ILT pore opening remains coupling to late voltage sensor movements and is detected by extracellular fluorescence measurements similar to those employed in this manuscript (doi: 10.1085/jgp.200409197). The fact that a fluorescence component is not observed with pore opening of KCNQ2*-R214W could suggests a different mechanism of decoupling than ILT, or that the late component of gating charge movement associated with pore opening is not reported by the fluor.

We thank the reviewer #1 for pointing out this difference that was also brought up by Reviewer#2. We agree that the decoupling mechanism induced by KCNQ2-R214W seems to be different from that seen with the ILT mutation in Shaker and from the KCNQ1-F351A mutant, as also pointed out by reviewer#2. We have performed more experiments with the R214W mutant and did not see a second fluorescence component. Therefore, in the revised version we have removed from Results: “The separation between F(V) and G(V) suggests that, like the uncoupling mutation F351A in KCNQ1 channels(36, 37, 45) or the ILT mutation in the Shaker K^+^ channels(46), R214W dissociates voltage sensor movement from channel opening.”

In the revised version of Result we simply say that: “The separation between the F(V) and G(V) curves suggest that R214W dissociates voltage sensor (S4) movement from channel opening. We hypothesize that, since R214W is in the loop connecting S4 to the S4-5 linker (not within the voltage sensor itself, Figure 5-figure supplement 2A), it most likely affects activation gating without directly affecting S4 movement.”

Specifically, we discuss these mechanisms in pages 14-15 of the revised manuscript. “…Unlike the ILT mutation in the Shaker K^+^ channels, in which pore opening is dissociated from the first VSD activation but coupled to the second (54), KCNQ2*-R214W fluorescence signals do not show a second fluorescence component associated with channel opening. This suggests that the R214W and the ILT mutants decouple VSD-PD pore through different mechanisms, or alternatively through similar mechanisms but our labeled F192C is unable to resolve the late component of gating charge movement (fluorescence) associated with pore opening.

Fig 6B Something seems wrong here: the fit is purported to represent 1/alpha, but with zalpha = 0.43 the tau fast at +40 should be 2.7x faster than -20 mV, and this is not the case.Fig 6C Tau on F slow as well as Tau on G slow (Fig 3 C) seem to lose their voltage dependence at more positive voltages. This could mean pore opening itself (gamma) has little voltage dependence.Fig 6F Why connote that f is always to the 1st power? Do L and f only impact the opening rate? The modeling is not described sufficiently to reproduce it. alpha and beta are missing altogether. zdelta/zgamma should also be listed in Supplement Table 2Page 11 "We find no experimental evidence supporting a constitutive open state (O0 in Figure 6F, gray)" From the model you can calculate the expected Popen at very negative voltages, it could be just a very low Popen.Page 12 "By decreasing the opening transition (L in Figure 6F) relative to wt KCNQ2 channels, the model also describes the clear separation between F(V) and G(V) curves observed in mutated KCNQ2*-R214W channels (Figure 6H), under the assumption that R214W changes the voltage sensing domain-pore domain (VSD-PD) coupling such that it prevents opening before multiple S4 have activated (Figure 6F, dashed maroon arrow)."Difficult to parse this sentence. In addition to L, f and gamma are also changed. What changes VSD-PD coupling, isn't that L?Page 12 "Additionally, data from the R198Q mutation can also be simulated by shifting the voltage dependence to negative voltages. " I imagine this is probably right but the claim is not justified by simulations.Page 13 Confidence in estimated total gating charge of 7.37 e0 per channel moved during KCNQ2 activation gating is limited due to the issue poorly described parameterization of the gating model.

The above 8 points and concerns raised by reviewer#1 are related to the Markov kinetic model shown in Figure 6 from the original manuscript. As stated above in *Recommendations for the Authors*, this model has been removed from the revised version of the manuscript, as also suggested by all three reviewers.

Page 14 "the overall voltage-dependent gating mechanisms of KCNQ2 is qualitatively similar to that of KCNQ1". To me it seems that the overall voltage-dependent gating mechanism of KCNQ2 is qualitatively distinct from KCNQ1.

This statement was primarily inferred from the kinetic model and the Rb/K experiments presented in the original submission, which have been removed in the revised version. Our data shows that both the steady-state voltage dependence of S4 transitions and the kinetics closely follow those of ionic currents. This result, in part, lead us to conclude that the gating of KCNQ2 resembles that of KCNQ1 (although we do not show evidence of intermediate open state, which remains to be tested in future studies) and KCNQ3 channels, but differed from homologous KCNQ4 channels. In addition, whether KCNQ2 channels need one (like KCNQ1) or multiple S4 movements to open requires additional support and further experimentation in future studies. We have revised and considered the reviewer’s comment in the discussion of this resubmitted manuscript (page 13).

Reviewer #2 (Recommendations for the authors):1. In Abstract and other places, the sentences such as "channel opening does not require multiple VSD movements" are not clear. Do the authors try to say, "the movements of multiple VSDs" or "the movements of VSDs in multiple steps"?

Thank you for this important observation, the wording we used was clumsy. Since we removed the kinetic model (Figure 6 in the original manuscript), we have also deleted any sentences that discuss concerted or independent S4 movement in the Abstract and Result sections. We only discussed that these alternatives, concerted or independent S4 movement, might explain our VCF data which shows that both the steady-state voltage dependence of S4 transitions and the kinetics closely follow those of ionic currents. Both references – Osteen et al PNAS 2010 and Westhoff et al PNAS 2019 have also been added – as recommended by the reviewer and apologize for overlooking these references in the original manuscript.

2. Did N190C really modify the channel at both -80 mV and +20 mV? Why would the same covalent modification of the channel at two voltages result in different GV relations (Fig 1E)? Is there another Cys in the channel that was modified differently at the two voltages? Did the authors use different protocols to measure GV in these two conditions? These need to be explained. The authors claimed a similar result in a reference, but it was not obvious if the reference showed the similar result or explained the result. The authors can cite the reasons given by the reference (40) if these can make sense of the results in Fig 1E.

Thank you for pointing out this important point. We have spent a good deal of time since we received the reviews answering this important point that was also raised as a concern by Revewer# 1. To that end, we have included additional data that support the idea that N190C channels are accessible in both the open and closed states. This is now clearly addressed in *Recommendations for the Authors, first Specific Suggestions* from Reviewer #1. See above Response to the first Specific suggestions from Reviewer# 1 on Pages 2-5.

In the original submission, we only used the protocols shown old Figure 1. We applied MTSET only at +20-mV for the open state and – 80-mV for the closed state. We used – 100-mV and – 120 mV for the closed state of A193C and S199C, respectively, because compared to the wt channels, these cysteine mutants shifted the GV relationship to negative voltages.

In the revised version, to further strengthen our conclusions, we have used a new protocol: For each cysteine mutant, we have designed a protocol in which we first apply MTSET at hyperpolarized voltages (closed) before switching to depolarized voltages (open) on the same cell, in a pairwise manner.

This is now described in the Result subsection “State-dependent external S4 modifications consistent with S4 as voltage sensor”, Pages 6-8 of the revised manuscript and new Figure 1 and Figure 1-figures supplement 3 and 4.

We also apologize for the lack of clarity in citing reference 40 in the original submission. This reference is deleted in the revised version, in light of our new data on N190C (new Figure 1 and Figure 1-figures supplement 3 and 4), which strengthen our claims that N190C modification occurs in in both states (open and closed).

3. Fig 1J at -80 mV: why is the current amplitude 0? It does not seem to be consistent with the description in the legend or J'.

We apologize for this confusion. The amplitude of the current 400 ms after the start of the + 20-mV voltage step (dashed vertical arrows in old Figure 1J’ and J’’) is not zero. What it is zero is the delta current measured between the first sweep (before MTSET, sweep #0) and the subsequent sweeps (after several MTSET application at – 80 mV, which are represented by #1, 2, …9).

In the revised version, we have now changed the Y axes as “Normalized D current”, and corrected the new Figure 1E and I as follow:

(I) The rate of MTSET modification of R198C channels at + 20-mV (red squares) or – 80-mV (gray squares) was measured using the difference in current amplitudes (taken at 400 ms after the start of the +20-mV voltage step, vertical dashed arrows in E) between the first sweep (before MTSET application, which is represented by #0 along the vertical dashed arrows in E, and normalized to zero) and the subsequent sweeps (after several MTSET application, which are represented by #1, 2…8-9 along the vertical dashed arrows in E) from the ‘closed- and open- state’-middle panels. The normalized delta current amplitude was plotted versus the cumulative MTSET exposure and fitted with an exponential. The fitted second-order rate constant in the open state protocol is shown in red. kopen = 3,230 ± 3.8 M–1 s–1 (*n = 3*).

4. Fig 2C and Fig 2-FigS1E: The FV with Alexa labeling increases at voltages >0 mV. Is this real or artifact? If real, does it indicate a second VSD movement?

We agree with the reviewer that the claim of a lack of an intermediate open state in KCNQ2 channels based on the Rb/K data provided in the original submission assumed that the pore properties of KCNQ2 are the same as those seen in KCNQ1 channels. Since we did not show sufficient experimental evidence to prove this point, we have removed Figure 6 (the model) from the revised manuscript. In the future, we will provide more evidence to build stronger support for the potential existence of intermediate and active open states in KCNQ2 channels. As such, we have removed the model shown in the original manuscript. Future studies will be performed to refine the KCNQ2 model, including the use of mutations that can lock the S4 in the intermediate or activated states in KCNQ2, as has been performed in the KCNQ1 channel by Zaydman et al; (PMID: 25535795). These experiments will provide more conclusive results regarding the different S4 states.

We have now re-analyzed the data and concluded that while the time course of the fluorescence appeared to have multiple exponentials, our fluorescence data lacked sufficient resolution to reliably estimate the first (fast) component. This might be because of the low signal-to-noise ratio of our VCF or/and because the filtering might have limited the tau-on from the optical signal (shown to be 20 ms in Figure 3C of the original submission). Please, also see above response to Reviewer #1, page 6.

As suggested by reviewers # 3, we have removed the kinetics comparison of fluorescence and current in the revised version of Figure 3, and simply state: …” There is a close correlation between the time course of fluorescence signals and ionic currents at all the voltages tested (Figure 3B, D). The close correlations in time (Figure 3) and voltage dependences (Figure 2G) of S4 motion (fluorescence) and activation gate (ionic current) resemble those observed for homologous KCNQ1 (without KCNE1)(42) and KCNQ3 channels(41, 43).”

As for the last part of the reviewer comments, the apparent increase in fluorescence after a plateau at voltages > 0mV has now also been revised. We have attempted new VCF at voltages more positive than + 40 mV to probe if a putative second fluorescence component after the plateau phase develops or if it is just artifacts of the experimental system. To get reliably fluorescence signals, we need a huge expression of labeled KCNQ2* channels (often producing currents larger than 100uA). It is considerably more difficult to properly clamp these high expressing cells, especially at extreme voltages. This experimental limitation makes it challenging to draw conclusions about the occurrence of a second fluorescent component. It may be possible to perform the cut—open technique coupled with VCF in order to shed light on this issue, but these experiments would require significant upgrade of the set up that we currently do not have this in place.

5. In Fig 4E, the comparison between MTS modification rate of A193C and FV of KCNQ2* makes no sense. These two curves derived from different mutations and modifications may overlap by coincidence.

We thank the reviewer for this comment. However, we believe that the comparison is still useful. To add more context explaining the rationale of this comparison, in the revised version (Pages 9-10) we now state: …“Because steady-state conductance/voltage curve, G(V), of A193C and labeled F192C channels are similar (G_1/2_ A193C = – 70 ± 2.4 mV, (n = 12) and G_1/2_ F192C_-Alexa_ = –77.1 ± 2.7 mV, (n = 9), Figure 4B and B’), and under the assumption that these two channels use the same S4 movement to generate these similar G(V)s, we will compare the voltage dependence of the modification rate (mod. rate) of A193C with the voltage dependence of the fluorescence of labeled F192C…”

Similar comparisons between the voltage dependence of modification rate by MTSET and S4 movements (by fluorescence of gating currents) have been previously used to independently assess the voltage range of S4 movement in different channels (PMID 9655514: Fig. 5A and PMID: 24769622: Fig 2C-D).

6. The data in Fig 6B-E seem to differ from the data in Fig 3C,D although the legend of Fig 6 claims that these are the same data. For instance, in Fig 3C the fast component of F does not show a voltage dependence, but in Fig 6B and C the fast and slow components of F show a similar voltage dependence.7. The model in Fig 6F raises several concerns: (1) Why do the vertical transitions have the rates of VSD activation, while they should represent pore opening/closing? (2) What does f represent in the scheme? Can it be part of L? (3) Detailed balance is violated in the left-most loop connecting C0, O0, O1, and C1.8. In Fig 6G, which states were used to represent currents? Which states represent fluorescence? Particularly, with both the horizontal and vertical transitions represent VSD activation, the rationale for simulating fluorescence need to be justified and the methods clearly described.

The reviewer raises an important concern in our original Figure 6F (model). Based on the Editors and reviewers comments, we have removed Figure 6 from the original manuscript to eliminate any of potential misunderstanding about the data presented. In future studies, we will gather additional fluorescence and current data using different protocols and dimer constructs to provide a more in depth description of KCNQ2 gating.

Reviewer #3 (Recommendations for the authors):I was very happy to read this paper and feel that this work is an important step forward for those working on these channels. A lot of progress has been made with KCNQ1 because of the relative ease of recording VCF signals, whereas similar work in KCNQ2-5 has been difficult. The KCNQ2-5 channels differ significantly from KCNQ1 in terms of their function and auxiliary protein regulation, so the development of useful tools to carry out detailed biophysical studies on these channels is valuable and took quite a heroic effort.I have quite a few comments, just important things that I would add to the paper in the interest of being thorough and not over-interpreting some of the findings.1. Page 3. "muscarine-regulated M-current". I would hesitate to call it 'muscarine regulated' as it can be sensitive to a variety of neurotransmitters that signal via Gq (ie. acetylcholine... I agree with the historical perspective of naming the current, but the wording may imply physiological regulation to some readers).2. Page 9: "These data also suggest that an individual voltage sensor movement might be sufficient to open the channel". The basis for this interpretation is not clear (at least not at this stage of the paper).

Thank you for the helpful comment. We agree that this was brought up very early in the paper and without robust support by the current set of data. This concern was also shared by Reviewers #1 and 2. We have revised the abstract and Results sections to more clearly describe the results presented. In the revised version, we only discuss that either concerted or independent S4 movement, might explain our VCF data which shows that both the steady-state voltage dependence of S4 transitions and the kinetics closely follow that of ionic currents.

Subsequent studies using one, two, or three concatemeric constructs containing mutations that prevent voltage sensors from moving into activated conformations, like those shown in Westhoff M et al., PMID: 30918124, could shed light on the concerted or independent nature of S4 movement in KCNQ2 channels in the future.

Also, as mentioned in the public review, a counterpoint to this observation is that the KCNQ2 or KCNQ3 currents typically exhibit a sigmoidal time course (also see Figure 1) to activation which might be accounted for by a requirement for multiple subunits to reach an activated conformation. Could this arise because arise because the dye labeling may prevent complete VSD deactivation or interfere with gating in some other way. This is also brought up at the top of page 14 and I have concerns that this could be a contentious statement. I would suggest more caution when describing and interpreting these properties.3. One way to potentially address this explicitly (ie. point #2) would be to include a direct comparison of unmodified and modifier I192C (and maybe WT KCNQ2 as well) in Figure 2. It is 100% fine with me that there are differences, but it should be clearly shown and described as a consideration when interpreting data, and some comparison like that would help readers.

We appreciate this suggestion. We agree with the review on this point. We have now modified the language used in both the Result and Discussion section of the revised manuscript. This is now also addressed in detail in response to the public review above. Please, see also new Figure 2 and Figure 2-figure supplement 1.

We now state on page 8 of the revised manuscript:

…“We find that this region exhibits sensitivity to cysteine mutations (Figure 2B-D), similar to a previous report for homologous cysteine mutations in KCNQ3 channels(41). Compared to wt-KCNQ2 channels, the mutants Q188C, G189C, and N190C shift the steady-state conductance/voltage curve, G(V), towards positive voltages (ΔGV_1/2_ = + 9.3 ± 0.7 mV, ΔGV_1/2_ = + 24.2 ± 0.7 mV, and ΔGV_1/2_ = + 29.8 ± 0.3 mV, respectively), whereas the mutants V191C and F192C shift the G(V) curves towards negative voltages (ΔGV_1/2_ = – 2 mV ± 0.9 mV and ΔGV_1/2_ = – 12.5 mV ± 1.7 mV, respectively) (Figure 2C-D, open symbols and supplement Table 1). Unlike the F192C mutant, the wt channels and the other cysteine mutants exhibit a sigmoidal time constant of activation that appear to have multiple exponential components (Figure 2B), with the F192C mutant generating the fastest time course of current activation (Figure 2-figure supplement 1A). Moreover, all five cysteine substitutions showed a further leftward G(V) shift upon fluorophore labeling (Figure 2D, filled symbols). The mechanisms by which the cysteine substitutions and their dye-conjugated versions may alter some of the gating properties are unknown and were not investigated further”.

And later on Page 9:

…”The gating properties of KCNQ2* channels (G(V) and kinetics) deviate from that of wt and unlabeled KCNQ2-F192C channels (Figure 2G-I). Labeling F192C with Alexa488-maleimide (or with Dylight488maleimide) shifts the G(V) relationship to negative voltages relative to unlabeled KCNQ2-F192C and wt channels (ΔGV_1/2_ = – 21.3 ± 0.8 mV and ΔGV_1/2_ = – 35.4 ± 2.2 mV, respectively, Figure 2C, D, G and Figure 2-figure supplement 1D). Moreover, compared to wt KCNQ2 channels, both labeled and unlabeled F192C accelerate the time course of current activation (Figure 2H-I)”.

Please, see also discussion of these issues on Pages 12-13 of the revised manuscript.

4. The other technical concern that I had was about fitting the fluorescence traces and perhaps adding complexity where it is not needed and not necessarily supported by data (perhaps this is being done due to analogy to prior work in KCNQ1). Based on the sample sweeps, there does not usually seem to be a great reason to fit with 2 components (eg. Figure 3) - is it really necessary in this case (ie. would it make a difference in terms of the predicted currents, especially given the uncertainty about sigmoid character of current activation)? A few other issues with the description of the model are that some parameters appear to be missing from Supplemental Figure 2 (ie. Alpha and Beta rates, and z for gamma and delta rates). In the text it seems that the gamma and delta rates are meant to be associated with channel opening, but the large amplitude fast component (alpha+beta rates) seem to correlate with the early stages of channel opening, it seems. Perhaps clarifying this by showing individual fit components, or simplifying the fitting/model would be helpful.

We agree with the reviewer’s comment. In terms of the predicted currents and considering the gating changes induced by both the F192C mutation and the dye conjugation together with the low resolution of reliable estimates of the first component, it might not be necessary, nor appropriate, to fit with two exponentials.

As also suggested by reviewer#1, we have removed the kinetics comparison of fluorescence and current in the revised version of Figure 3, and simply state: …” There is a close correlation between the time course of fluorescence signals and ionic currents at all the voltages tested (Figure 3B, D). The close correlations in time (Figure 3) and voltage dependences (Figure 2G) of S4 motion (fluorescence) and activation gate (ionic current) resemble those observed for homologous KCNQ1 (without KCNE1)(42) and KCNQ3 channels(41, 43).”

As for the last part of the reviewer comments, we have removed the kinetic model as suggested by all three reviewers. We agree that the model presented in Figure 6 of the original submission was underdeveloped. We will aim to gathering more fluorescence and current data using additional approaches and concatemeric constructs to provide a more in-depth description of KCNQ2 gating in the future.